# UPF1/SMG7-dependent microRNA-mediated gene regulation

Jungyun Park [1,5], Jwa-Won Seo [2,5], Narae Ahn [1], Seokju Park [2], Jungwook Hwang [1,3] & Jin-Wu Nam [2,4]

The stability and quality of metazoan mRNAs are under microRNA (miRNA)-mediated and nonsense-mediated control. Although UPF1, a core mediator of nonsense-mediated mRNA decay (NMD), mediates the decay of target mRNA in a 3′UTR-length-dependent manner, the detailed mechanism remains unclear. Here, we suggest that 3′UTR-length-dependent mRNA decay is not mediated by nonsense mRNAs but rather by miRNAs that downregulate target mRNAs via Ago-associated UPF1/SMG7. Global analyses of mRNAs in response to UPF1 RNA interference in miRNA-deficient cells reveal that 3′UTR-length-dependent mRNA decay by UPF1 requires canonical miRNA targeting. The destabilization of miRNA targets is accomplished by the combination of Ago2 and UPF1/SMG7, which may recruit the CCR4-NOT deadenylase complex. Indeed, loss of the SMG7-deadenylase complex interaction increases the levels of transcripts regulated by UPF1-SMG7. This UPF1/SMG7-dependent miRNA-mediated mRNA decay pathway may enable miRNA targeting to become more predictable and expand the miRNA-mRNA regulatory network.

[1] Graduate School for Biomedical Science & Engineering, Hanyang University, Seoul, Republic of Korea. [2] Department of Life Science, College of Natural Sciences, Hanyang University, Seoul, Republic of Korea. [3] Department of Medical Genetics, College of Medicine, Hanyang University, Seoul, Republic of Korea. [4] Research Institute for Convergence of Basic Sciences, Hanyang University, Seoul, Republic of Korea. [5] These authors contributed equally: Jungyun Park, Jwa-Won Seo. Correspondence and requests for materials should be addressed to J.H. (email: jwhwang@hanyang.ac.kr) or to J.-W.N. (email: jwnam@hanyang.ac.kr)

The 3′ untranslated regions (UTRs) of messenger RNAs (mRNAs) are core noncoding structures that control the stability, localization, and translation of mature mRNAs at the posttranscriptional level[1,2]. The 3′UTRs of tissue-specific genes vary in length across different tissues more than those of housekeeping genes[3–5]. Changes made to 3′UTRs by alternative polyadenylation often lead to the inclusion and exclusion of *cis*-acting elements that interact with diverse RNA-binding proteins (RBPs) on 3′UTRs, thus altering the stability, localization, and translational regulation of corresponding mRNAs[6,7].

The most widely studied *cis*-acting elements in the 3′UTRs, which are microRNA (miRNA) recognition elements (MREs) that interact with miRNA-loaded Argonaute (Ago), regulate the stability and translation of target mRNAs. 3′UTR shortening by alternative polyadenylation excludes MREs, resulting in the avoidance of miRNA targeting; however, 3′UTR lengthening leads to the inclusion of more MREs, resulting in further downregulation of target genes[8–11]. The global shortening of 3′ UTRs has been observed in highly proliferating cells, activated T cells, and cancer cells, and the corresponding mRNAs are generally derepressed by avoiding miRNA targeting[8,9,12]. In contrast, globally progressive lengthening of 3′UTRs is observed during brain development[13], and extended 3′UTRs embed thousands of conserved MREs in mammals[10], strengthening miRNA-mediated gene regulation[14].

In addition to these *cis*-acting element-specific mechanisms, lengthening of the 3′UTR may influence the stability of mRNAs through the nonsense-mediated mRNA decay (NMD) pathway[15]. NMD is typically initiated by an aberrant stop codon at least 50–55 nt upstream of the last exon junction, known as a premature stop codon (PTC)[16–19]. Because this process is transduced by the exon junction complex (EJC) downstream of the PTC[20,21], it is also known as EJC-dependent NMD. EJC-independent NMD has also been reported to destabilize some mRNAs with a normal stop codon by modulating the distance between the poly(A)-binding protein (PABP) complex and the stop codon[22,23]. Hence, the lengthening of 3′UTRs can either trigger EJC-dependent NMD by introducing downstream exon junctions (dEJs) into the extended region, changing a normal stop codon to a PTC, or inducing EJC-independent NMD. EJC-dependent NMD targets are generally decayed through three distinct UPF1-dependent pathways after PTC recognition: (i) SMG6-mediated endoribonucleolytic cleavage near the PTC[24,25], (ii) SMG5-mediated decapping[26,27], and (iii) SMG7-mediated deadenylation of targets[28]. However, the detailed mechanism of EJC-independent NMD in the regulation of long 3′UTR transcripts remains unknown.

Previous studies have shown that UPF1 in association with 3′ UTRs senses the length of 3′UTRs to potentiate mRNA decay[29,30]. Global analysis of UPF1-dependent mRNAs (UMD) using UPF1 RNA interference (RNAi) and cross-linking immunoprecipitation (IP) followed by sequencing (CLIP-seq) has revealed that thousands of mRNAs are modulated in response to UPF1 knockdown and that hundreds of mRNAs with no annotated dEJs respond to UPF1 knockdown as the length of the 3′ UTR increases[31]. Motif analysis in the 3′UTR of target mRNAs revealed a high-GC context with a CUG-rich motif, which is required for UPF1-dependent mRNA decay[32]. This 3′UTR length-dependent regulation of UMD may expand the range of post-transcriptional gene regulation, although the specific mechanism and whether the regulation occurs through EJC-dependent or EJC-independent NMD are largely unclear.

In this study, we demonstrate that the 3′UTR length-dependent regulation of UPF1-dependent mRNA decay occurs through EJC-independent but miRNA-dependent regulation. This UPF1- and miRNA-dependent mRNA decay represses Ago2-UPF1-binding targets, which do not require TNRC6, suggesting that this mRNA decay pathway can be considered an alternative miRNA targeting pathway.

## Results

**UPF1-dependent, EJC-independent mRNA decay.** Although early studies have reported that a considerable fraction of RNAs responsive to UPF1 depletion are dependent on the 3′UTR length, the question of whether UPF1-responsive RNAs are degraded in an EJC-independent manner has repeatedly been asked. To address this unanswered question, we sought to filter EJC-dependent NMD target genes out and reanalyze the genes remaining (considered dEJ-free genes) to determine whether they continued to exhibit responses in UPF1-depleted cells (Fig. 1a; for additional details, see "Filtration of EJC-dependent NMD targets" in the Methods section). For dEJ-free genes, the changes in the expression of the genes binned based on their 3′UTR lengths were reinvestigated using publicly available RNA-seq data from UPF1-depleted human cell lines and mES cells[31,33–35]. If a gene had multiple transcripts (mRNAs), the transcript with the longest 3′ UTR was selected as the representative mRNA of the gene to measure the 3′UTR length. The bin sizes of the 3′UTR lengths were chosen to include an equal number of genes in each bin. Of the genes, 48.0–53.4% for HeLa cells, 37.1% for K562 cells, and 42.0% for mES cells were derepressed by a greater than $\log_2$ fold-change (0.2) in response to UPF1 knockdown (Supplementary Table 1). As a result, the 3′UTR-length-dependent expression changes in response to UPF1 knockdown remained significant in mES, HeLa, and K562 cells (Fig. 1b, c and Supplementary Fig. 1a, b) and were not dependent on the bin sizes of the 3′UTR lengths (Supplementary Fig. 1c).

To confirm the results in our system, RNA-seq experiments in siRNA for UPF1 (siUPF1)- and control siRNA (siControl; scrambled-siRNA)-treated HeLa cells were performed. Using the RNA-seq data and the pipeline shown in Fig. 1a, we compiled 7117 dEJ-free genes, 3419 (48.0%) of which were derepressed by a greater than $\log_2$ fold change (0.2) upon the downregulation of UPF1 (Supplementary Table 1), while 18.3% were downregulated by a less than $\log_2$ fold change (−0.2). The 3′UTR length-dependent regulation of genes was similar in our experiment (Fig. 1d). Taken together, these results indicate that UMD does not result from the EJC-dependent NMD pathway but rather from the EJC-independent NMD pathway in mammals.

**The 3′UTRs of UMD targets commonly contain MREs with CUG-rich content.** The lengthening or shortening of mRNA 3′ UTRs allows corresponding transcripts to contain more or less *cis*-acting elements by chance[8,9,11], leading to further regulation. Thus, we hypothesized that AU-rich elements (AREs; AUUA), GU-rich elements (GREs; UGUUUGUUUGU), and/or MREs are involved in regulating UMD targets, as they are comparatively more prevalent than others in the 3′UTRs. To identify the probable *cis*-acting elements, motif-enrichment analysis was performed with all 7-mers of the 3′UTR sequences of 3419 UMD targets showing a $\log_2$ fold-change ≥ 0.2 against dinucleotide-shuffled 3′UTR sequences ($P < 0.005$; Fisher's exact test). Highly ranked 7-mers generally showed a high GC content (Fig. 2a). GC-rich sequences tend to contain more stable secondary structures, which are preferred binding sites of UPF1 (Supplementary Fig. 2a)[31,35], and the UMD targets contained significantly more CUG motifs and extended 7-mers, CCUG[AG][AG][AG] (Fig. 2a), as previously described[32].

Among the significantly enriched 7-mers, 15.2% were 7-mer MREs of 682 human miRNA families[36] (Supplementary Data 1), with significantly higher levels than there would be by chance

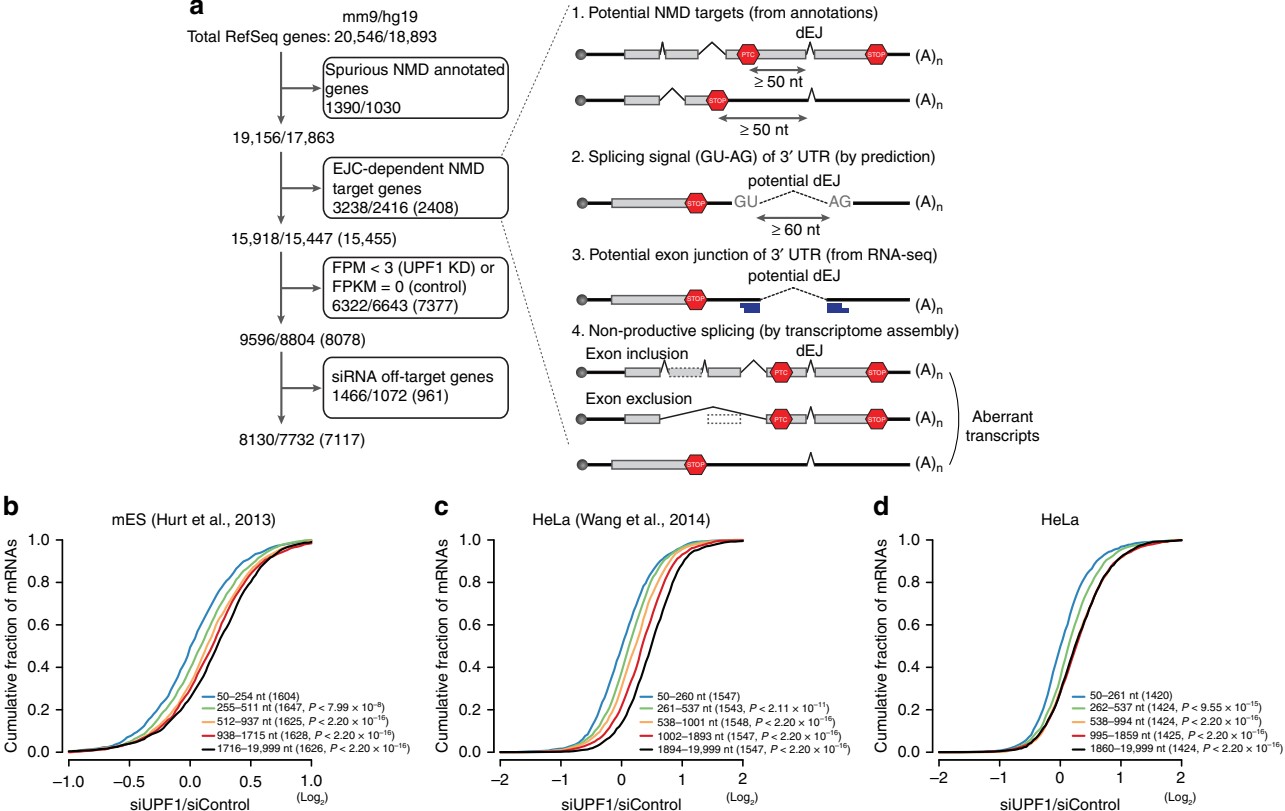

**Fig. 1** UPF1-responded genes. **a** Shown is the schematic flow for selecting dEJ-free genes, comprising four filtering steps (see "dEJ-free genes" in the Methods section for additional details). The top slash-delimited numbers indicate the RefSeq genes for humans (hg19) and mice (mm9). The following slash-delimited numbers indicate genes remaining from each indicated step, and the numbers in boxes indicate genes excluded at each indicated step. From the third step, only expressed genes were considered by quantifying their expression values (FPMs and FPKMs) with mES RNA-seq data[31], with HeLa RNA-seq data[35] and our experiment (parenthesis), in order. Red stop sign, stop codon in potential NMD targets; paired thick blue line, RNA-seq-supported exon junction read; dotted line, potential exon junction; dotted box, exon exclusion; dark-gray solid box, exon inclusion. **b–d** Cumulative fractions of log2 changes in the expression of genes in siUPF1-treated cells against siControl-treated cells are shown as cumulative distribution function (CDF) graphs of Hurt et al.'s mES cell data (**b**), Wang et al.'s HeLa cell data (**c**), and our HeLa cell data (**d**). The numbers in parentheses are dEJ-free genes without off-target sites of siRNAs belonging to each 3′UTR length bin. The P values were estimated by the Kolmogorov–Smirnov (K–S) test

(8.6%; $P < 2.2 \times 10^{-16}$; Fisher's exact test); 43 were 7-mer MREs of the 50 most abundant miRNA seed families, and 10 were 7-mer MREs of the 10 most abundant miRNA seed families ([37] for HeLa cells; Supplementary Table 2), which were significantly overrepresented (Fig. 2a; $P < 1.73 \times 10^{-8}$ for the top 50; $P < 0.0016$ for the top 10; Fisher's exact test with 1000 random miRNA families). The most significant MRE was the 7-mer site of miR-16-5p (Fig. 2a; $P < 1.99 \times 10^{-63}$; Fisher's exact test). In contrast, neither AREs nor GREs were detected.

Because of the MRE- and CUG-rich contexts in the 3′UTRs of the UMD targets, we examined whether CUG motifs are spatially related to MREs. To address this, we counted all CUG motifs near the 7-mer MREs of the 50 most abundant miRNA seed families in the 3′UTR of the UMD targets. The CUG motif was significantly enriched in the 5-nt upstream and downstream windows of the MREs compared to randomized controls ($P < 2.20 \times 10^{-16}$; Fisher's exact test; Fig. 2b and Supplementary Fig. 2b). Although not as much as that of the UMD targets, the CUG motif of the non-UMD targets was also significantly enriched in the windows compared to the randomized control ($P < 2.20 \times 10^{-16}$; Fig. 2b). We thus hypothesized that enrichment of the CUG motif could result from enrichment of the CAG (base-pairing to the CUG) motif in seed miRNAs. In fact, the signal-to-noise ratios of all 3-mers in the 3′UTR of the UMD targets (Supplementary Table 3) were significantly correlated with those of all 3-mers in the

sequences, reverse complementary to all annotated human miRNAs (Supplementary Table 3) (Fig. 2c; Pearson's correlation $r = 0.71$; $P < 5.90 \times 10^{-11}$). CUG was one of the most significantly enriched 3-mers in both sequences (Fig. 2c); similarly, CUG was conserved in mammalian, fly, and worm miRNAs (Supplementary Tables 3–5). Notably, 12 (24%) of the 50 most abundant miRNA seed families and 144 (13.8%) of all annotated human miRNA families[36] with embedding of the CAG triple-nucleotide base-pairing to 7-mer sites of the UMD targets were significantly more enriched than those of random sequences (Fisher's exact test; Fig. 2d, e).

**UMD requires miRNA-mediated gene regulation.** The enrichment of CUG triplet nucleotides in miRNA 7-mer sites raised the question of whether UMD is related to miRNA-mediated gene regulation. Changes in the expression of the dEJ-free mRNAs with 7-mer sites of the 50 most abundant miRNA families in HeLa and mES cells (Supplementary Table 2) were measured between siUPF1-transfected and siControl-transfected cells and compared to those of 3′UTR-length-controlled genes with random 7-mer sites (Supplementary Fig. 3a; see "Generation of random controls and statistical test" in the Methods section for more details). The miRNA targets with 7-mer sites were significantly more derepressed than were controls in both public and

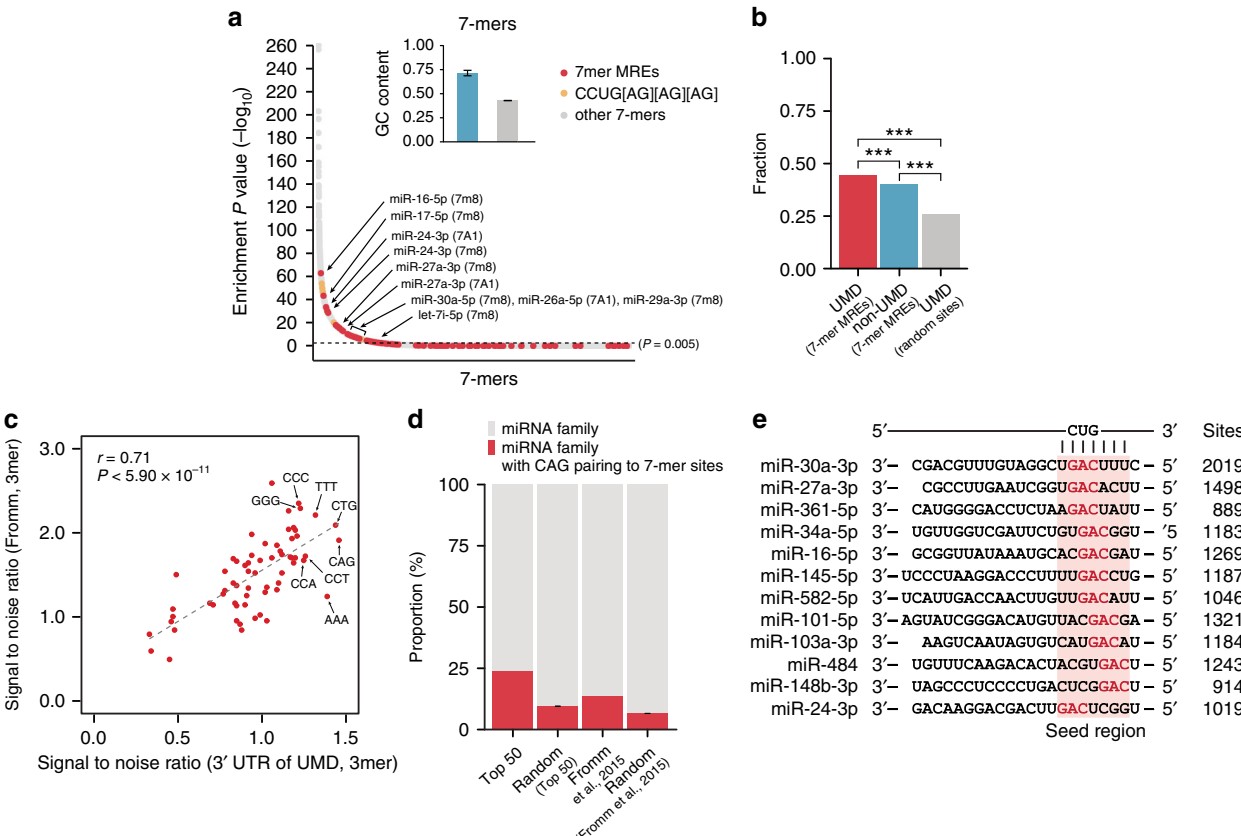

**Fig. 2** Co-occurrence of the CUG motif and MREs in UMD targets. **a** Analysis of all possible 7-mers enriched in the 3′UTR of UMD targets. 7-mer MREs of the 50 most abundant miRNA families (red dot), eight CCUG[AG][AG][AG] motifs (orange dot), and other 7-mers (gray dot) are shown with enrichment P values (nominal). The black dotted line is a threshold for significant enrichment (nominal P value < 0.005). The indicated 7-mer MREs are the sites of the 10 most abundant miRNA families. The site in the parentheses indicates the type of miRNA 7-mer sites. The inset bar graphs show the GC contents of other significant 7-mers (cyan box; more significant 7-mers than the 7-mer site for miR-16-5p) and less or nonsignificant others (gray box; less significant 7-mers than the 7-mer site for miR-16-5p). The error bars indicate the standard errors of the mean (s.e.m.) of the GC contents. **b** The counts of CUG motifs within a ±5-nt window of a thousand 7-mer MREs in the 3′UTRs of UMD targets (red) were compared to those within the ±5-nt window of a thousand 7-mer MREs in the 3′UTRs of non-UMD targets (cyan) and those within the ±5-nt window of a thousand 7-mer random sites in the 3′UTR of UMD targets (gray). Asterisks denote the statistical significance of the differences (***$P < 2.20 \times 10^{-16}$). **c** The signal-to-noise ratios of 3-mers from sequences reverse complementary to annotated human miRNAs (x-axis) and from the 3′UTR of UMD targets (y-axis) are shown in an x–y plot. The dashed line is fitted to the data with Pearson's correlation coefficient ($r = 0.71$). **d** The proportion of the 50 most abundant miRNA families (in HeLa) and their dinucleotide-shuffled random sequences (100 cohorts) and all human miRNA families and their dinucleotide-shuffled random sequences (100 cohorts) that embed CAG pairing to their 7-mer sites. The error bars indicate the s.e.m. of the proportions for random miRNAs. **e** Of the 50 most abundant miRNA families, 12 embedded CAG motifs in their seed regions (pink box). Sites are the number of miRNA 7-mer sites in the 3′UTRs of UMD targets

our RNA-seq data (Fig. 3a; $P \le 0.014$ for all three; one sample t test), although the level of miRNAs was not significantly changed in siUPF1-transfected cells (Supplementary Fig. 3b). Even for the 10 and 30 most abundant miRNA families in HeLa cells, the targets containing the sites were significantly more derepressed than in the controls (Fig. 3a), suggesting that miRNA-mediated gene regulation is likely involved in UMD.

We then examined whether the derepression of UMD targets in response to UPF1 knockdown was diminished in the absence of miRNAs. HeLa cells were transfected with siUPF1, siRNA for Dicer1 (siDicer1), siUPF1/siDicer1, or siControl, and the level of UMD targets was assessed by RNA-seq. Western blot analysis indicated that UPF1 and Dicer1 were efficiently downregulated compared to the endogenous protein Calnexin (Fig. 3b). Indeed, the depletion (Supplementary Fig. 3c) or knockout[38,39] of Dicer1 led to the global downregulation of miRNAs, resulting in the derepression of miRNA targets (Supplementary Fig. 3d)[38,40]. Changes in the expression of dEJ-free genes in siUPF1-treated cells were compared to those in siControl-treated cells with a Dicer1-depleted background, and the dysregulation of UMD

targets and their 3′UTR length-dependent repression disappeared in cells with the Dicer1-depleted background (Fig. 3c, d).

We next predicted that the number of MREs could explain the effect of the UMD targets according to their 3′UTR lengths under UPF1-depleted conditions because the 7-mer sites of miRNAs were increased by chance as the 3′UTRs lengthened (Fig. 3e). To test this hypothesis, changes in the expression of dEJ-free genes with different numbers of 7-mer sites were compared to controls with random sites. 3′UTR length-dependent regulation resulted from the increase in MREs (Fig. 3f and Supplementary Fig. 3e, f), which depended on the presence of Dicer1 (Fig. 3g), suggesting that UMD requires miRNA targeting. In fact, the derepression of UMD targets by UPF1 depletion was explained by a computational model (wContext++ model of TargetScan 7.0[41]) of miRNA targeting (Supplementary Fig. 3g, h, i).

**UMD is activated via miRNA seed-type targeting.** Because of the requirement for miRNA targeting in UMD, we evaluated how UPF1 and Ago2 function together to determine the fate of

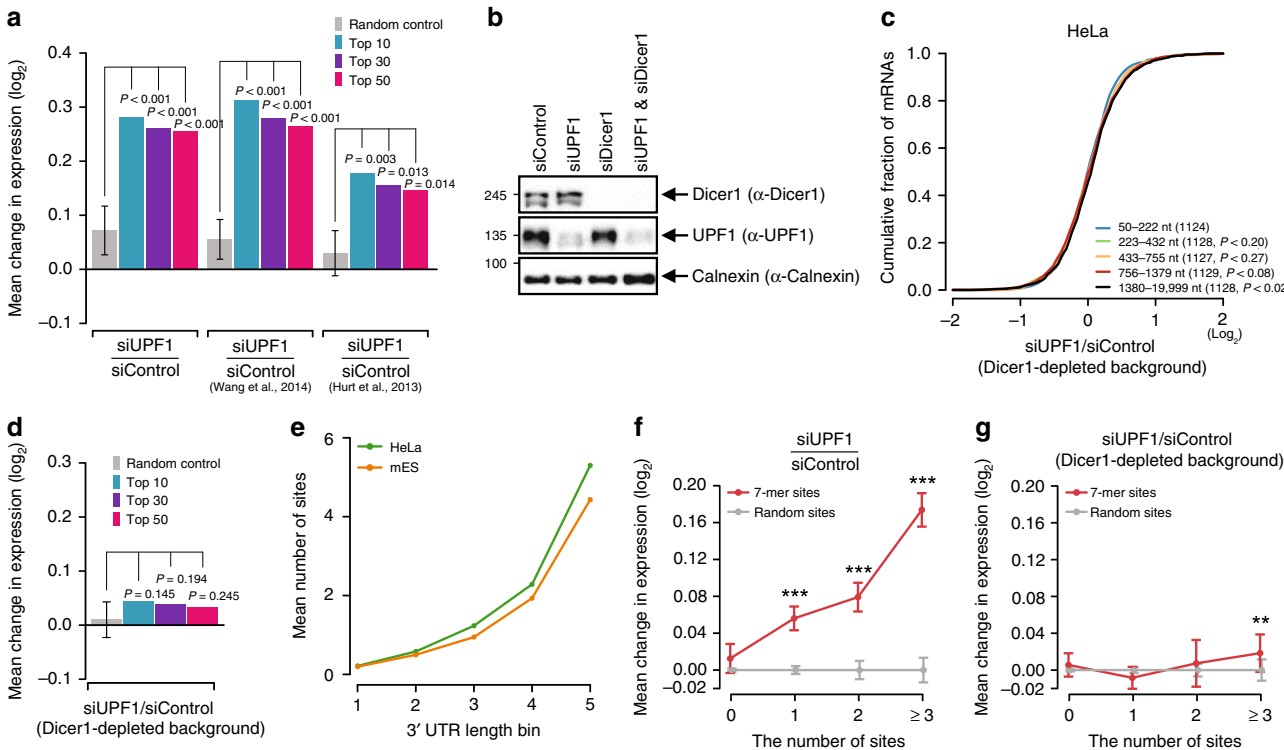

**Fig. 3** UMD regulation requires miRNAs. **a** The mean changes in expression (log$_2$ scale) of dEJ-free mRNAs embedding 7-mer target sites of random controls (gray), the 10 (blue), 30 (purple), and 50 (red) most abundant miRNA families under siUPF1-treated conditions are shown. The $P$ value was calculated by comparing the mean change in the expression of the targets to those in the 1000 random control cohorts (see "Generation of random controls and statistical tests") using a one sample $t$ test. The same analyses were performed for our data (left), Wang et al. 2014's data (middle), and Hurt et al.'s data (right). The error bars indicate the s.e.m. in the changes for the targets. **b** HeLa cells were transfected with siUPF1, siDicer1, or both siUPF1 and siDicer1. The downregulation of endogenous UPF1 and/or Dicer1 was confirmed by western blotting. Calnexin served as a loading control. **c** CDF graphs of changes in the expression of dEJ-free mRNAs between siUPF-treated and siControl-treated cells under Dicer1-depleted conditions are shown across different 3′UTR length bins. Otherwise, as in Fig. 1b–d. **d** Mean changes in the expression of dEJ-free mRNAs with 7-mer target sites between siUPF-treated and siControl-treated cells under Dicer1-depleted conditions. Otherwise, as in **a**. **e** Mean numbers of 7-mer miRNA target sites in the 3′UTRs of dEJ-free mRNAs are shown across different 3′UTR length bins. The sites were detected in the major 3′UTR isoform, as evident by our previous 3P-seq data profiled from HeLa (green) and mES (orange) cells[11]. **f** Mean changes in expression of dEJ-free mRNAs in siUPF1-treated cells against siControl-treated cells are shown across the different 7-mer site numbers of the ten most abundant miRNA families (red) and different random 7-mer sites (gray). Asterisks denote groups whose number of sites is significantly different from that of the controls (***$P < 0.001$; K–S test). **g** Mean changes in expression between dEJ-free mRNAs of siUPF-treated and siControl-treated cells under Dicer1-depleted conditions are shown for the different 7-mer site numbers. Otherwise, as in **f**

mRNAs by analyzing public UPF1 and Ago2 CLIP-seq data[42,43]. As previously reported[31], UPF1- and Ago2-binding sites significantly overlap compared to randomized controls in the 3′UTR (Supplementary Fig. 4a, b; $P < 3.26 \times 10^{-6}$ for HeLa cells; $P < 9.52 \times 10^{-7}$ for mES cells; Fisher's exact test). They appeared to be proximally colocalized with each other (Supplementary Fig. 4c, d). To examine whether colocalization of Ago2 and UPF1 on miRNA target sites is necessary for miRNA-dependent UMD, we analyzed changes in expression of the targets with Ago2-UPF1 overlapping sites. The changes in the expression of the targets with an Ago2-UPF1 overlapping site (for the 50 most abundant miRNAs) were marginally greater than those of the targets with a single Ago2 site (Supplementary Fig. 4e; $P = 0.07$; Wilcoxon's rank-sum test).

To experimentally validate the effect of the Ago2–UPF1 interaction on the UMD targets, we first examined miR-24-3p targets with 7-mer sites because miR-24-3p was found to be one of the ten most abundant miRNA families, and its 7-mer sites were significantly enriched in the UMD targets (Fig. 2a). Of the miR-24-3p targets upregulated (a log$_2$ fold-change greater than 0.3) in UPF1-knockdown cells compared to control cells and greater than those in Dicer1-depleted cells (yellow-colored shades in Fig. 4a),

78 with a wContext + + score less than −0.3 were considered confident miRNA-dependent UMD targets (green-colored circles in Fig. 4a). Among the 78 targets, nine UPF1-dependent miR-24-3p targets (*DNAL4*, *ATAD2B*, *RPS19BP1*, *PEA15*, *CRAT*, *ELL*, *PAK4*, *DBNDD1*, and *ZNF740*) were selected using the following criteria in order of priority: (1) targets more derepressed in UPF1-knockdown cells than in control cells; (2) targets with miR-24-3p 8-mer or 7m8 sites but no sites for the other nine most abundant miRNAs; and (3) a representative target in each 3′UTR length bin. Reverse transcription quantitative polymerase chain reaction (RT-qPCR) using the lysate of HeLa cells transfected with siUPF1 was carried out for the nine selected targets, eight of which were consistently derepressed in the cells (Fig. 4b). We then examined whether the selected targets were actual miRNA targets via 7-mer sites. The levels of UPF1-dependent miR-24-3p targets containing 7-mer sites on the 3′UTR were quantified from the lysate of cells transfected with an miR-24-3p inhibitor (antisense) and/or an miR-24-3p pseudo-mimic (Fig. 4c). All nine UPF1-dependent miR-24-3p targets, a UPF1-independent miR-24-3p target (*HPCAL1*) and two known miR-24-3p targets (*MXI1* and *ATG4A*)[44,45] were effectively derepressed by introducing the miR-24-3p inhibitor. The miR-24-3p pseudo-mimic restored the

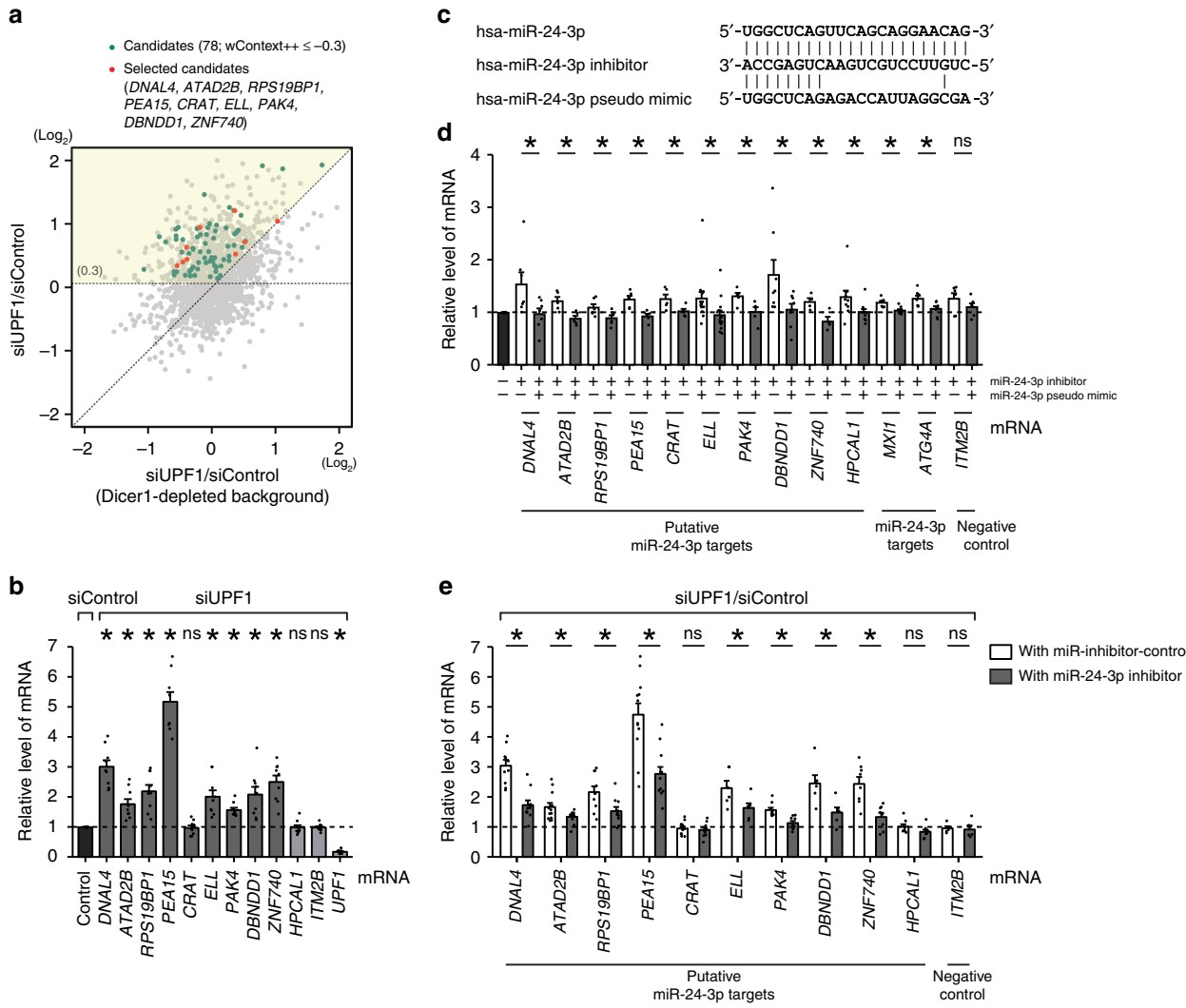

**Fig. 4** Experimental validation of miRNA-dependent UMD targets. **a** Comparisons of changes in the expression (log$_2$ scale) of mRNAs with 7-mer sites in response to UPF1 depletion in siDicer1-treated and WT backgrounds. The yellow-shaded area indicates miRNA-dependent UMD targets. Candidates with a wContext + + score of less than −0.3 were selected (green, red dots), nine of which were chosen for experimental validation (red dot). **b** The increased level of the selected candidates obtained from **a** was confirmed by RT-qPCR. The level of mRNA was normalized to that of *GAPDH* mRNA. **c** Sequences of human miR-24-3p, the miR-24-3p antisense inhibitor, and the miR-24-3p pseudo-mimic used in this study, where the sequence of the pseudo-mimic was identical to that of the miR-24-3p seed sequence but with different sequences beginning at the ninth nucleotide. A pseudo-mimic was used because pseudo-mimics can form base pairs with the 7-mer sites of targets but avoid inhibition by miRNA inhibitors. **d** HeLa cells were transfected with the inhibitor and/or pseudo-mimic, and RT-qPCR was performed to quantify the level of putative and known miR-24-3p targets. *ITM2B* mRNA, which does not contain any miRNA-binding sites, served as a negative control. The level of mRNA was normalized to that of *GAPDH* mRNA. **e** HeLa cells transfected with siUPF1 or siControl under miR-24-3p-depleted or control conditions. RT-qPCR was performed to quantify the level of transcripts. Mean values and s.e.m. were calculated from independent experiments. An asterisk denotes statistically significant differences (*$P < 0.05$; unpaired Student's $t$ test); ns not significant. The minimum number of independent biological replicate experiments was **b** $n = 9$, **d** $n \geq 4$, and **e** $n \geq 5$

effects of the inhibitor, suggesting that decay of the targets was regulated through the 7-mer site of miR-24-3p (Fig. 4d). To verify that UMD is activated via miR-24-3p-mediated targeting, we examined the effects of UPF1 knockdown under miR-24-3p-depleted conditions (Fig. 4e). RT-qPCR results demonstrated that all putative miR-24-3p targets except for *CRAT* mRNA showed greater derepression in the presence of miR-24-3p (treated with miR-inhibitor-control) than in the absence of miR-24-3p (treated with miR-24-3p inhibitor) in response to UPF1 knockdown (Fig. 4e; Supplementary Fig. 4f); this is presumably because impaired miRNA targeting diminishes the effect of UMD. The above results were also observed for another miRNA, miR-26a lacking a CAG motif, one of the ten most abundant miRNA

families, as depicted in Fig. 2e (Supplementary Fig. 4g–k). Taken together, our findings suggest that UMD functions via miRNA seed-type targeting.

**UMD functions through the CUG motif on MREs.** To assess the role of CUG motifs in the miRNA-mediated silencing of UMD targets, changes in the expression of dEJ-free genes with 7-mer sites harboring the CUG motif of the 50 most abundant miRNA families were compared to those of genes with neither 7-mer sites nor CUG motifs, to those of genes with only CUG motifs, and to those with both but not overlapping in their 3′UTRs. Although targets containing only the CUG motif were slightly derepressed

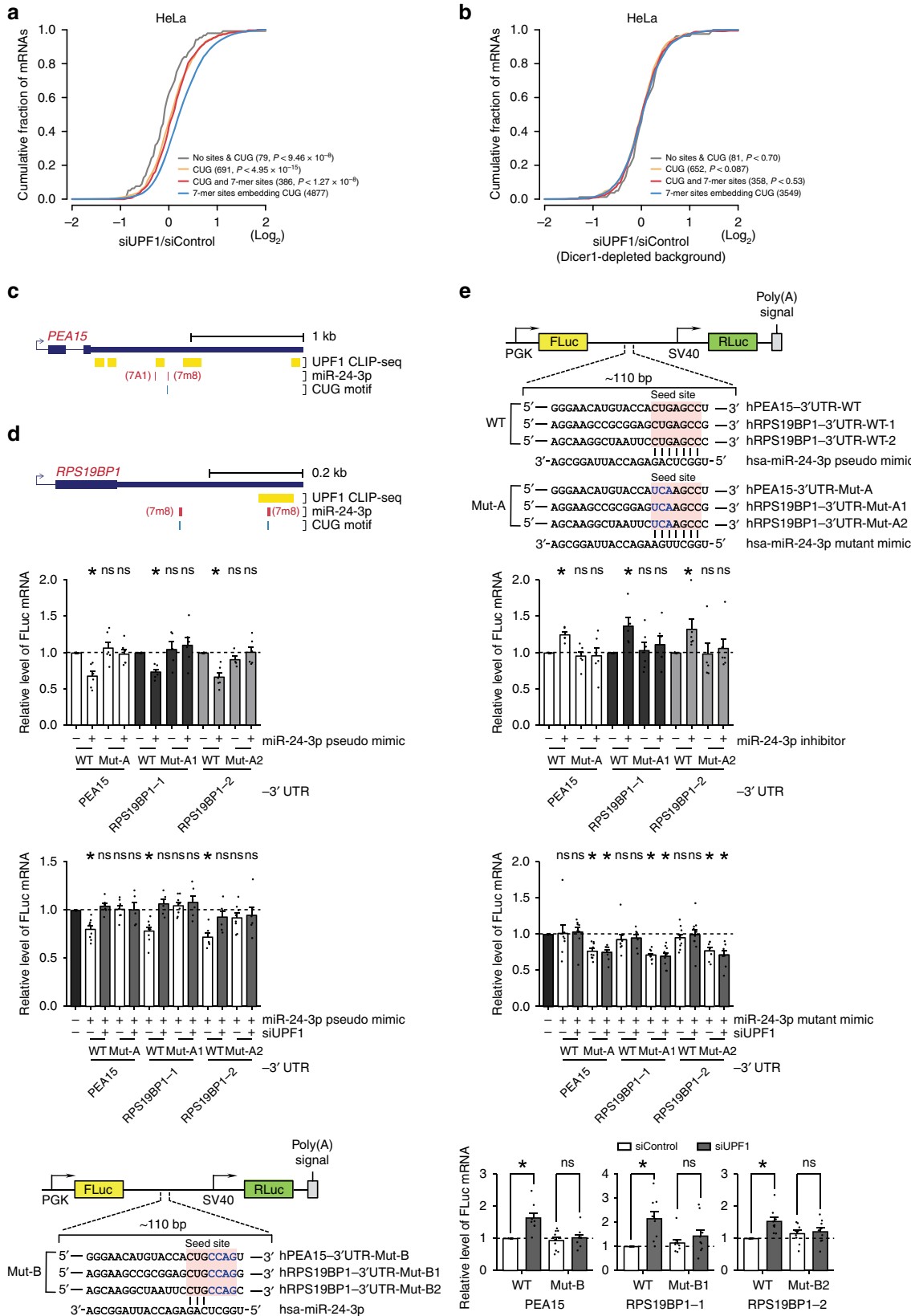

compared to those with neither sites nor CUG motifs (Fig. 5a; $P < 0.02$; K–S test), expression of UMD target genes with 7-mer sites with an embedded CUG motif responded more significantly than did those with only a CUG motif (Fig. 5a; $P < 4.95 \times 10^{-15}$; K–S test) or those with both in the 3′UTRs (not CUG-embedded)

with UPF1 knockdown ($P < 1.27 \times 10^{-8}$; K–S test). However, the effects of CUG embedded in 7-mer sites disappeared under Dicer1-depleted conditions (Fig. 5b).

To validate the effect of UPF1-binding CUG motifs embedded in 7-mer sites of UMD targets, we chose two transcripts, *PEA15*

**Fig. 5** miRNA-dependent UMD is activated via CUG motifs. **a**, **b** CDF graphs of changes in the expression of UMD target mRNAs, including CUG alone in the 3′UTR (orange), both CUG and 7-mer sites not overlapping with each other within the ±15 nt window (red), and CUG-embedded 7-mer sites (blue), were compared to those embedding neither 7-mer sites nor the CUG in 3′UTRs (gray) in siUPF1-treated versus siControl-treated HeLa cells (**a**) and siUPF1-treated versus siControl-treated HeLa cells under Dicer1-depleted conditions (**b**). Otherwise, as in Fig. 1b–d. **c**, **d** (3′UTRs of *PEA15* (**c**) and *RPS19BP1* (**d**) that include UPF1-binding sites, miR-24-3p 7-mer sites, and CUG motifs. **e** Schematic representation of bicistronic Firefly(FLuc)/Renilla(RLuc) luciferase plasmid (reporter constructs) containing the putative miR-24-3p-binding sequence (red box, seed site) from *PEA15* or *RPS19BP1* 3′UTR. The CUG motif (black bold) in the seed site was mutated to UCA (blue bold). The hsa-miR-24-3p mutant mimic sequence has a complementary sequence to the mutated sequence in the seed region. **f–i** HeLa cells were cotransfected with reporter constructs and the miR-24-3p pseudo-mimic (**f**), miR-24-3p inhibitor (**g**), miR-24-3p pseudo-mimic and siUPF1 (**h**), miR-24-3p mutant mimic and siUPF1 (**i**), and siUPF1 (**j**). Control siRNA was transfected rather than siUPF1 as a control in **h** and **i**. RT-qPCR was performed to measure the relative amount of FLuc mRNA. Mean values and s.e.m. were calculated from independent experiments. (*$P < 0.05$; unpaired Student's *t* test); ns not significant. The minimum number of independent biological replicate experiments was **f** $n \geq 4$, **g**, **h** $n \geq 5$, **i** $n \geq 7$, and **j** $n = 9$

(1882 nt of 3′UTR) and *RPS19BP1* (395 nt of 3′UTR), which contain putative miR-24-3p-binding site(s), and showed a greater response to UPF1 depletion (Fig. 5c, d). The putative miR-24-3p-binding sites in the 3′UTR were inserted into the reporter constructs (Fig. 5e). Furthermore, the CUG motif in the seed region was mutated to UCA (Mut-A), which prevents miR-24-3p from binding to the 3′UTR of the reporter constructs. As expected, the miR-24-3p pseudo-mimic reduced the levels of wild-type (WT) reporter transcripts but not those of mutant (Mut-A) transcripts (Fig. 5f; Supplementary Fig. 5, top). In contrast, introducing a miR-24-3p inhibitor (Fig. 5g; Supplementary Fig. 5, bottom) and downregulating UPF1 (Fig. 5h) derepressed the levels of WT reporter transcripts but not those of mutants (Mut-A). To examine whether UMD depends on the CUG motif embedded in the 7-mer site, the miR-24-3p mutant mimic, which is capable of binding to Mut-A reporter transcripts but not to WT reporter transcripts, was introduced in the presence or absence of siUPF1 (Fig. 5i). The mutant mimic reduced the level of Mut-A reporter transcripts, which were not recovered by downregulation of UPF1, indicating that UMD depends on the CUG motif embedded in the 7-mer site of miR-24-3p. Because a miRNA inhibitor may not completely remove miR-24-3p, the UPF1-independent function via the CUG motif might also explain our RT-qPCR results. To exclude the possibility of the UPF1-independent function via the CUG motif, we mutated the 7-mer site with the remaining CUG (CUG<u>AGCC</u> to CUG<u>CCAG</u>) in the reporter transcripts (Mut-B). The level of the Mut-B transcripts was not affected by downregulating UPF1 (Fig. 5j), suggesting that UMD triggers mRNA decay via miRNA-mediated gene regulation in a CUG-motif-dependent manner and that the cooperative roles of Ago2 and UPF1 may be necessary for UMD.

**UPF1 helicase activity, SMG6, and SMG5 are not related to UMD**. We next asked how the Ago2–UPF1 axis mediates UMD. UPF1 functions as an indispensable helicase for NMD and removes NMD complexes after the cleavage of mRNA by SMG6[46,47]. Based on this, we hypothesized that helicase activity could enhance miRNA targeting by either increasing the target accessibility of structured sites in the 3′UTR or by cleaving miRNA targets with SMG6. To test this hypothesis, we ectopically expressed a UPF1 helicase mutant (R843C) in HeLa cells after downregulating endogenous UPF1 by RNAi. Western blot analysis indicated that the expression levels of UPF1-WT and UPF1-R843C were comparable, while that of endogenous UPF1 was low after UPF1-WT or UPF1-R843C transfection (Supplementary Fig. 6a). RNA-seq of WT and UPF1-R843C mutant HeLa cells was performed, and changes in expression of dEJ-free genes were examined. In addition, the changes in expression were examined using publicly available RNA-seq data from mES cells treated with siRNA for SMG6 (siSMG6)[48]. Changes in expression,

however, were not observed in mRNAs with different 3′UTR lengths in cells expressing UPF1-R843C (Supplementary Fig. 6b) or in siSMG6-treated mES cells (Supplementary Fig. 6c), indicating that the miRNA-mediated mRNA decay enhanced by UPF1 resulted from neither improved target accessibility nor the direct cleavage of targets by SMG6.

Phosphorylated UPF1 also interacts with SMG5 and SMG7, which control the deadenylation and decapping of NMD targets, respectively[28,49]. We thus examined whether SMG5 and/or SMG7 are involved in UMD. For this, we examined changes in dEJ-free gene expression depending on their 3′UTR length using publicly available microarray data profiled from HeLa cells treated with siRNA for SMG5 (siSMG5) or siRNA for SMG7 (siSMG7)[26]. siSMG5 treatment did not affect expression of these genes (Supplementary Fig. 6d), whereas the results of siSMG7 resembled those of siUPF1 (Fig. 6a). The results of siSMG7 were also confirmed in other publicly available RNA-seq data for SMG7-depleted cell lines[50], for which different siSMG7 sequences were employed (Supplementary Fig. 6e).

**SMG7 is considerably responsible for UMD**. To further examine the relationship between SMG7 and UMD targeting, we performed RNA-seq with total RNAs isolated from the lysates of scrambled siRNA-treated and siSMG7-treated HeLa cells and observed gradual derepression according to 3′UTR length (Fig. 6b; $P < 2.58 \times 10^{-12}$; K–S test) and significantly enriched 7-mer MREs and CUG-embedded motifs in the 3′UTRs of SMG7-dependent targets (Fig. 6c), similar to UPF1-dependent targets. Analysis of UMD targets with CUG-embedded 7-mer sites (Fig. 6b) or significant 7-mer sites (Supplementary Fig. 6f) revealed that the changes in expression were explained mostly by targeting the ten most abundant miRNA families (Fig. 6d) and correlated with the number of 7-mer sites (Fig. 6e), although the magnitude of the changes in expression was lower than that under UPF1 knockdown. In fact, the change in expression of the UMD targets with significant 7-mer sites in siSMG7-treated cells reached approximately 40% of that in siUPF1-treated cells (Figs. 3a, 6d, and Supplementary Fig. 6g).

To experimentally examine the functional interaction of UPF1 and SMG7 in UMD, we first evaluated miR-24-3p targets with 7-mer sites, which were derepressed in both siUPF1- and siSMG7-treated cells. Of the 1488 miR-24-3p targets with 7-mer sites in their 3′UTRs, 301 were depressed by more than a 0.2 log_2 fold-change in siSMG7-treated cells. Indeed, 187 of the targets were UPF1- and SMG7-dependent miR-24-3p targets. Of the nine UPF1-dependent miR-24-3p targets depicted in Fig. 4, eight were UPF1- and SMG7-dependent miR-24-3p targets, and the remaining one (*ZNF740*) was the only UPF1-dependent miR-24-3p target (Fig. 6f). To verify these candidates, the levels of candidates, including the SMG7-dependent miR-24-3p target *HPCAL1*, upon the depletion of UPF1 and/or SMG7 were

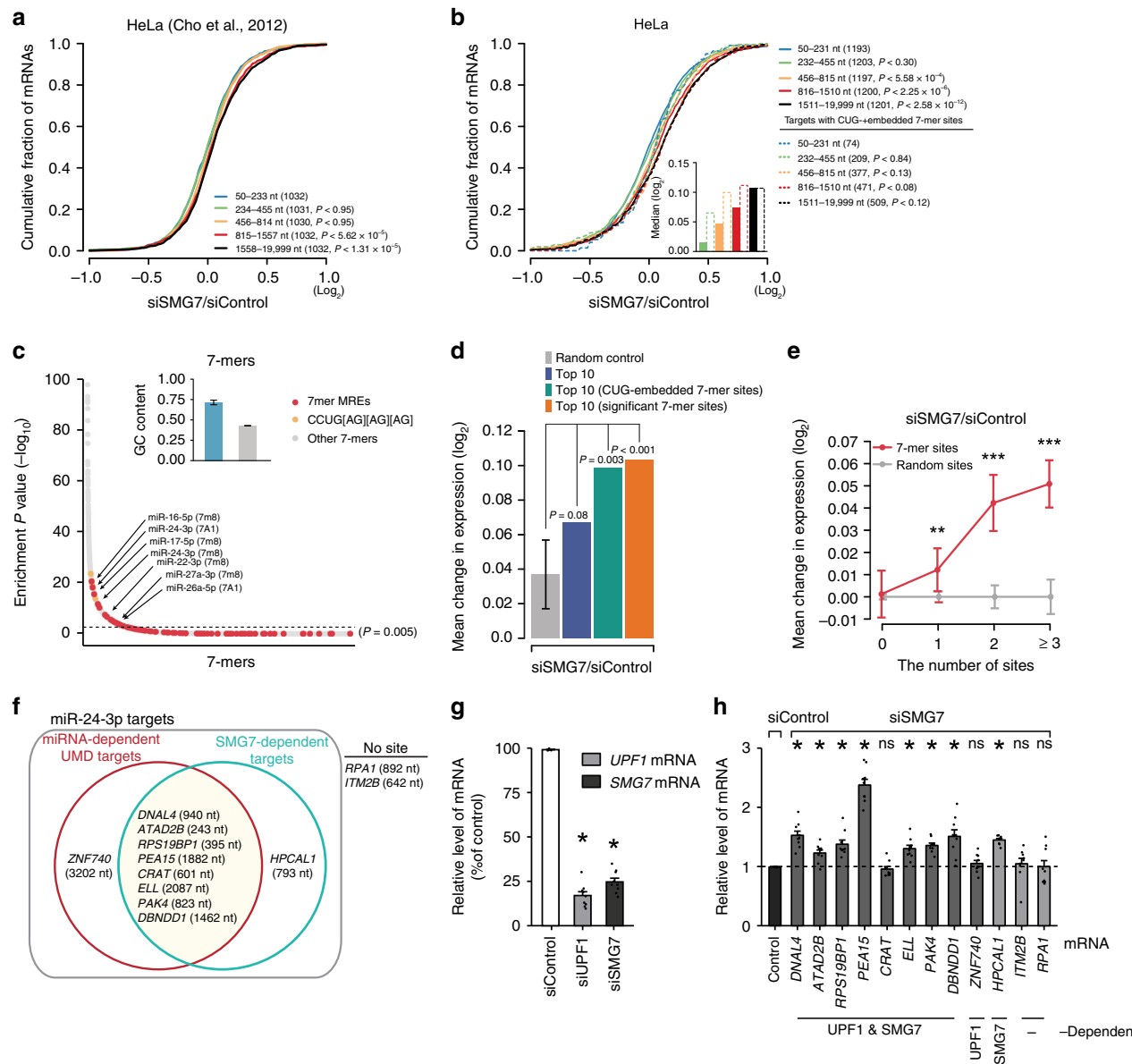

**Fig. 6** Subset of miRNA-dependent UMD targets depends on SMG7. **a** CDFs for changes in the expression of dEJ-free mRNAs in microarray data prepared from siSMG7-treated and siControl-treated cells are shown. Otherwise, as in Fig. 1b–d. **b** as in (**a**), CDF graphs of changes in the expression of dEJ-free mRNAs (solid lines) and targets with CUG-embedded 7-mer sites of the 50 most abundant miRNA families (dotted lines) in siSMG7-treated and siControl-treated cells. Median expression fold-changes are depicted by box plots (inset). Blank boxes with a dotted line denote the median fold-changes of targets with CUG-embedded 7-mer sites. **c** Analysis of all possible 7-mers enriched in the 3′UTR of SMG7-dependent mRNAs. Otherwise, as in Fig. 2a. **d** Mean changes in expression of a random control (gray), dEJ-free mRNAs with 7-mer sites (blue), targets with CUG-embedded 7-mer sites (green), or targets with significant 7-mer sites (orange) of the ten most abundant miRNA families in siSMG7-treated versus siControl-treated cells. Otherwise, as in Fig. 3a. **e** Mean changes in expression of dEJ-free mRNAs in SMG7-knockdown versus control cells are shown over different numbers of miRNA sites (red) or random sites. Asterisks denote statistically significant differences (**$P < 0.01$; ***$P < 0.001$; K–S test). Otherwise, as in Fig. 3f. **f** Venn diagram of nine selected miRNA-dependent UMD target candidates in Fig. 4a and nine SMG7-dependent targets. The values in parentheses are the lengths of 3′UTR in each transcript. "No site" indicates that the mRNAs did not contain any putative miRNA sites of the 50 most abundant miRNAs. **g** Endogenous UPF1 and SMG7 in HeLa cells were depleted by siRNA-mediated downregulation. RT-qPCR was performed to examine the efficiency of downregulation. The level of mRNA was normalized to that of GAPDH mRNA. **h** As in **g**; however, the levels of transcripts listed in **f** were examined by RT-qPCR. Mean values and s.e.m. were calculated from independent experiments. (*$P < 0.05$; unpaired Student's $t$ test); ns not significant. The minimum number of independent biological replicate experiments was $n = 9$ in **g**, **h**

quantified by RT-qPCR. The levels of endogenous UPF1 mRNA and SMG7 mRNA were effectively downregulated to approximately 30% (Fig. 6g). RT-qPCR of SMG7-depleted HeLa cell lysates indicated that eight SMG7-dependent miR-24-3p targets (including HPCAL1) were consistently derepressed in nine

candidate genes (Fig. 6h). These results were verified by miR-26a targets, indicating that the levels of FAM98A and HTATIP2 mRNA increased upon SMG7 depletion (Supplementary Fig. 6h). Considering these findings, SMG7 may play a role in miRNA-mediated UMD.

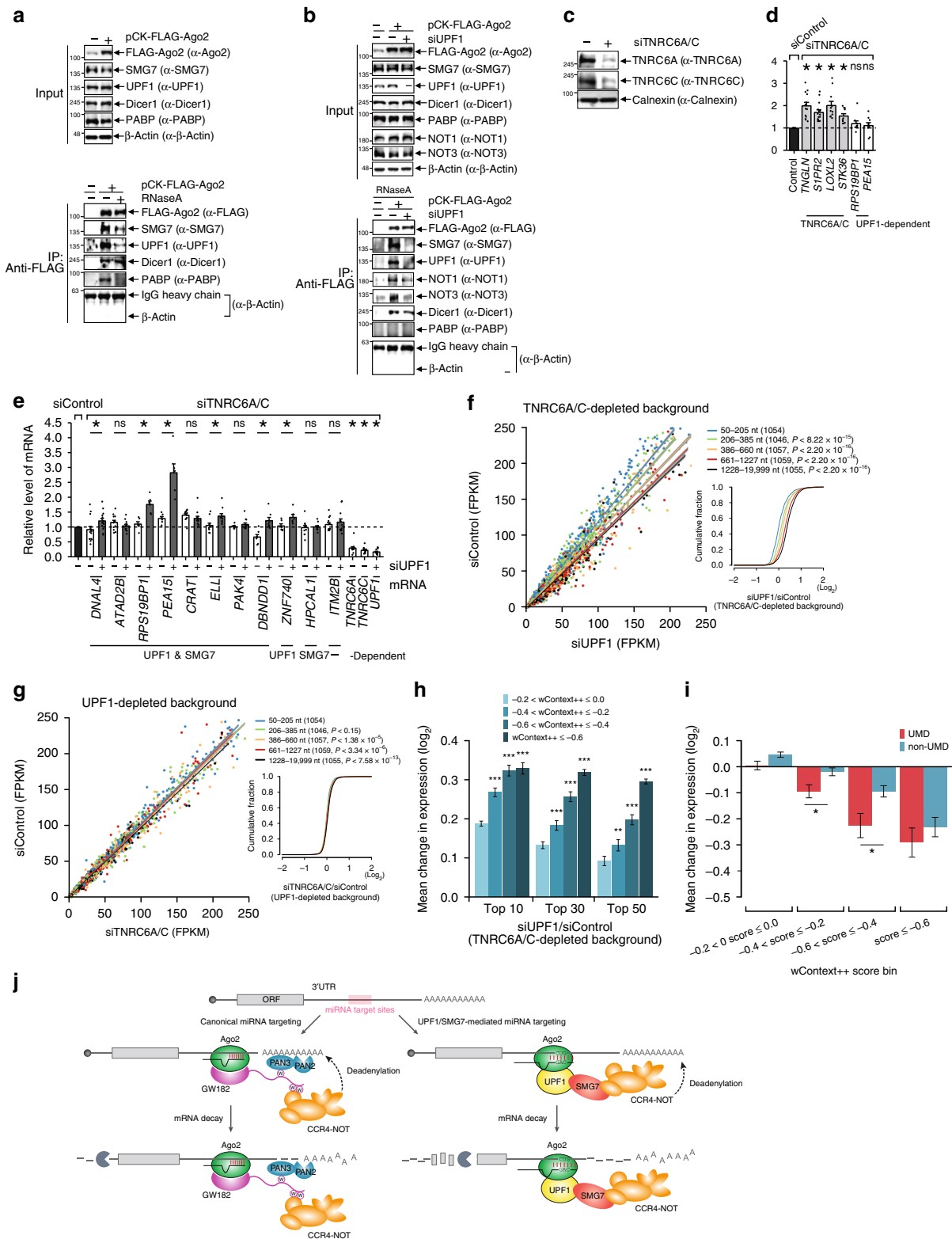

**UPF1-SMG7 forms a complex with Ago2.** To determine whether the UPF1–SMG7 complex is associated with the Ago2 complex, HeLa cells were transiently transfected with FLAG-Ago2 or empty vector containing FLAG-tag. Cell lysates were subjected to IP with anti-FLAG antibodies in the presence or absence of RNaseA treatment (Fig. 7a). Western blot analysis of the IP inputs and eluates indicated that FLAG-Ago2 was efficiently and comparably immunoprecipitated by anti-FLAG antibodies. The amounts of Dicer1 that coimmunoprecipitated with FLAG-Ago2 were comparable, while coimmunoprecipitated PABP was observed only in the absence of RNaseA treatment, indicating that RNA was efficiently degraded. As expected, UPF1 and SMG7 coimmunoprecipitated with FLAG-Ago2 in a partially RNA-independent manner, suggesting that UPF1–SMG7 could form a complex with Ago2. To determine whether SMG7 joins the Ago2 complex depending on UPF1, HeLa cells were transiently

**Fig. 7** Alternative miRNA-mediated regulation through the UPF1-SMG7 complex. **a** The lysates of HeLa cells that were transiently transfected with pCK-FLAG-Ago2 or empty vector were subjected to IP using anti-FLAG antibodies in the presence (+) or absence (−) of RNaseA. Western blotting was performed to detect proteins in IP inputs and eluates. **b** Similar to **a**; however, HeLa cells were cotransfected with pCK-FLAG-Ago2 or an empty vector and siRNA specific for UPF1 or siControl. IP was performed in the presence of RNase A. **c**, **d** The downregulated level of TNRC6A/C was confirmed by western blotting (**c**). The relative levels of transcripts, which were UPF1-independent but TNRC6A/C-dependent (*TNGLN*, *S1PR2*, *LOXL2*, and *STK36*), were quantified by RT-qPCR (**d**). **e** Changes in the expression of mRNAs with respect to siUPF1 treatment are shown in the siTNRC6A/C background. Transcript levels were quantified by RT-qPCR. The level of mRNA was normalized to that of *GAPDH* mRNA. Mean values and s.e.m. were calculated from independent experiments. Asterisks denote statistically significant differences (*$P < 0.05$; unpaired Student's *t* test); ns not significant. **f**, **g** Endogenous TNRC6A/C, UPF1 or both were depleted by siRNA transfection. Expression levels of dEJ-free mRNAs were compared between siControl and siUPF1-treated cells under TNRC6A/C-depleted conditions (**f**) and between siControl and siTNRC6A/C-treated cells under UPF1-depleted conditions (**g**). The cumulative fractions of the dEJ-free mRNAs are shown over the $\log_2$ fold-change compared to the initial values (**f**, **g**). Otherwise, as in Fig. 1b. **h** Mean changes in expression of dEJ-free mRNAs with 7-mer sites of the 10, 30, and 50 most abundant miRNA families in response to UPF1 depletion in the TNRC6A/C-depleted background are shown across different wContext + + score bins. Error bars are the s.e.m. of the changes in expression. **i** Mean changes in the expression of UMD (red) and non-UMD targets (blue) with miR-124 and miR-155 7-mer sites are shown over the different wContext + + bins. **j** Proposed miRNA-mediated UMD model and canonical miRNA-mediated gene silencing pathway. The minimum number of independent biological replicate experiments was $n \geq 9$ in **d** and $n \geq 6$ in **e**

transfected with FLAG-Ago2 one day after treatment with siControl or siUPF1 (Fig. 7b). UPF1 was efficiently downregulated, while other proteins were unchanged by transfection with siUPF1. In contrast with the co-IP of SMG7 with FLAG-Ago2 in the presence of UPF1, the level of SMG7 was not detectable with FLAG-Ago2 in UPF1-depleted cells, indicating that SMG7 forms a complex with Ago2 by interacting with UPF1. Furthermore, NOT1 and NOT3, which are components of the deadenylase complex, were coimmunoprecipitated with FLAG-Ago2, and the amounts of coimmunoprecipitated NOT1 and NOT3 were reduced by the downregulation of UPF1. To show whether the interaction of SMG7 and the deadenylase complex is involved in UMD, we employed a FLAG-tagged SMG7 mutant (FLAG-SMG7 ΔPC), which failed to interact with the deadenylase complex[28] (Supplementary Fig. 7a). The RT-qPCR results indicated that the loss of SMG7 interaction with the deadenylase complex stabilized the miR-24-3p targets (except *ATAD2B*) and was regulated by UPF1, but did not stabilize the negative controls (Supplementary Fig. 7b). These observations provide insight into the unknown possible mechanism of UMD, i.e., UPF1/SMG7-dependent miRNA targeting.

**UMD targeting appear not to require TNRC6A/C.** We then examined whether UPF1/SMG7-dependent miRNA targeting requires the GW182 complex, which interacts with the canonical CCR4/NOT pathway. Thus, we first downregulated two major paralogs of GW182, TNRC6A/C, using siRNA. Western blot analysis showed that endogenous TNRC6A/C were efficiently downregulated (Fig. 7c). The RT-qPCR results indicated that the TNRC6A/C-dependent and UPF1-independent transcripts were upregulated by TNRC6A/C depletion (Fig. 7d). Then, we downregulated UPF1 and/or TNRC6A/C using siRNA (siTNRC6A/C; Supplementary Table 6) and examined changes in the expression of UPF1-SMG7-dependent miR-24-3p targets, a UPF1-only-dependent miR-24-3p target and nontarget compared to control cells transfected with scrambled siRNA (Fig. 7e). Interestingly, no examined genes, including the UPF1-dependent miRNA targets, were significantly changed by the downregulation of TNRC6, except for *PEA15* and *CRAT*, which were slightly increased; these targets were derepressed by UPF1 depletion, except for *ATAD2B*, *CRAT*, and *PAK4* (Fig. 7e). These results suggest that the UPF1/SMG7-dependent miRNA targeting pathway might function in a TNRC6A/C-independent manner.

To validate whether UPF1-dependent miRNA targets function independently of TNRC6A/C, changes in expression of dEJ-free genes by UPF1 knockdown were compared with those of the control under TNRC6A/C-depleted background conditions.

The 3′UTR length dependency of dEJ-free genes was also observed by additional UPF1 knockdown under TNRC6A/C-depleted conditions (Fig. 7f), and changes in expression of UMD targets were mostly attributed to miRNA targeting (Supplementary Fig. 7c). In contrast, 3′UTR length-dependent changes in expression of dEJ-free genes were not observed by downregulating TNRC6A/C in a UPF1-depleted background (Fig. 7g), and the changes in expression of miRNA targets were smaller than those of the targets in a UPF1-knockdown cells with the siTNRC6A/C background (Supplementary Fig. 7c, d). miRNA targets with multiple sites, however, were more derepressed in the absence of TNRC6 as the number of sites increased (Supplementary Fig. 7e, f). RT-qPCR using TNRC6A/C-depleted HeLa cells was performed to quantify the mRNA levels of UPF1-SMG7-dependent miR-26a targets, UPF1-dependent targets, and non-targets compared to those in siControl-transfected cells (Supplementary Fig. 7g). As expected, none of the genes examined, including UMD targets, were significantly changed by the downregulation of TNRC6A/C, except for *GREB1L* and *HUWE1*, which were increased slightly (Supplementary Fig. 7g).

In addition, the responses of UPF1/SMG7-dependent miRNA targets with respect to siUPF1 treatment were more effectively explained by the wContext + + score sum of miRNAs in cells with a TNRC6A/C-depleted background (Fig. 7h), and UPF1/SMG7-dependent miRNA targets had a greater impact by ectopically expressing miRNAs in HeLa cells (Fig. 7i), but not siTNRC6A/C in cells with an UPF1-depleted background (Supplementary Fig. 8). Taken together, these results indicate that UMD targeting occurs through a distinct pathway involving the Ago2/UPF1/SMG7 axis rather than the canonical Ago2-TNRC6 miRNA targeting pathway, and miRNA target prediction can be more predictable through UMD targeting.

## Discussion

In this study, we propose a UPF1/SMG7-dependent miRNA-mediated mRNA decay pathway, which appeared to be the major molecular mechanism of UMD, through experimental and computational analyses in mammalian cells. The proposed miRNA targeting pathway indicates the presence of an alternative regulatory layer in miRNA-mediated gene silencing. UPF1-dependent miRNA targeting via the CUG motif may explain the evolutionarily conserved relationship between miRNA-mediated mRNA decay and the NMD pathway.

Although the series of RNAi experiments for UPF1 and SMG7 supported the existence of UMD targeting, the results of our RNAi experiments could suffer from the issue of off-target effects. However, even after excluding all off-targets with 6-mer

seed sites, our results and conclusions were not changed (Supplementary Fig. 9a, b), and repeated experiments using antisense oligonucleotides (ASOs), which depend on RNaseH cleavage, confirmed what we observed in our RNAi experiments (Supplementary Fig. 9c, d). These results suggest that our findings are not the result of off-target effects of siRNAs.

Both the candidate experiments and global analysis of gene expression changes revealed that UPF1-dependent miRNA targeting is mediated by the UPF1–SMG7 interaction. Although our IP results suggest interactions between Ago2 and UPF1, we cannot completely exclude the possibility that Ago2 indirectly interacts with UPF1 because the interaction may result from the proximal interaction of two proteins via either undigested RNA or other mediators. In addition, although the Ago2 and SMG7 interaction appears to be dependent on UPF1, the UPF1-SMG7 pathway can partly explain UPF1-dependent miRNA targeting (Fig. 6). Approximately 40% of the mean change in the expression of the miRNA targets (for the 50 most abundant miRNA families) under UPF1-depleted conditions, compared to that of the random control, was mediated by the UPF1-SMG7 pathway (Figs. 3a, 6d). This finding could indicate the presence of a mediator other than SMG7 or technical limitations, including limitations in knockdown efficiency or experimental variations.

CAG triplet nucleotides were overrepresented in the seed region of both highly expressed and low-abundance miRNAs in the cells of humans and other animals, including flies and worms (Fig. 2c and Supplementary Tables 3–5), suggesting that miRNAs nonrandomly possessed CAG triplet nucleotides in the seed regions and that miRNAs embedding the CAG motif were preferentially selected for targeting UMD targets during evolution. This finding also suggests that the UPF1-dependent, TNRC6-independent miRNA targeting pathway via the CUG motif does not occur in certain cell types but is a form of general, evolutionarily conserved gene regulation. Interestingly, the CUG motif was preferentially observed in UPF1-dependent targets and thought to be within the UPF1-binding motif. Because CAG triplet nucleotides in the seed regions of miRNAs are on the inside of Ago2[51], UPF1, and Ago2 cannot structurally bind to the CUG motif at the same time, suggesting that miRNA-loaded Ago2 transiently binds to UPF1 on the CUG motif in MREs, in turn allowing Ago2 to directly bind to the MREs (Fig. 7j). However, it is not clear why UPF1 does not block AGO binding to a site. These questions should be further investigated.

## Methods
**Data sources**. NCBI RefSeq gene annotation (hg19: Aug-22-2011, mm9: Feb-15-2014) was used throughout the analyses. To precisely measure the 3′UTR length and expression level of mRNAs, the end of the 3′UTRs was updated with the major form profiled from 3P-seq of HeLa and mES cells. RNA-seq data for UPF1-knockdown and WT cells were downloaded from GSE63091 and DRP000622 for HeLa cells, from GSE88140 and GSE88083 for K562 cells, and from GSE41785 for mES cells. CLIP-seq data for WT UPF1 in HeLa cells were from GSE47976 and in mES cells were from GSE41785. Raw CLIP-seq data for WT Ago2 of HeLa cells were from GSE43666, and the processed CLIP-seq data of mES cells were from GSE25310. 3P-seq data for HeLa cells and mES cells were from GSE52531. sRNA-seq data for HeLa cells were from GSE22068, and those for mES cells from GSE20384.

**Statistical tests**. All statistical tests were performed using one-tailed tests in R packages unless otherwise specified. For all CDF and miRNA targeting analyses, K–S tests and one-sample $t$ tests were performed unless otherwise specified. For enrichment analyses, Fisher's exact tests were carried out.

**Filtration of EJC-dependent NMD targets**. For this, we first removed known EJC-dependent NMD target genes and all genes that included any transcript with potential PTCs in the annotation (NCBI RefSeq: see "Data Source" in the Methods section for additional details). Because the annotations are far from complete and often lack aberrantly spliced transcripts of a gene, we additionally excluded genes with aberrantly spliced transcripts possibly including dEJ (defined as exon-exon

junction after 50 nucleotides downstream of a stop codon) by assembling transcripts from the RNA-seq data of UPF1-depleted cells using our computational pipeline (see "dEJ-free gene" and "Filtering genes with nonproductive splicing forms" in the Methods section). In addition, we removed all the siRNA off-targets (for UPF1, GFP, and the scrambled siRNAs) that included 7-mer sites in the 3′ UTR (see "Filtering siRNA off-targets" in the Methods section for additional details), and the remaining targets were considered dEJ-free genes (7732 for HeLa cells; 8,130 for mouse embryonic stem [mES] cells), which were used in ensuing analyses.

**dEJ-free genes**. All analyses started with RefSeq genes that transcribe nonspurious mRNAs with 5′UTR ≥ 25 nt, CDS ≥ 200 nt, and 3′ UTR ≥ 50 nt. The annotated (known) NMD target genes (if any transcript was an annotated NMD target) were first excluded from the RefSeq genes. All potential EJC-dependent NMD target genes, including putative dEJs (defined as an exon junction after 50 nt downstream from a stop codon), were then excluded. dEJs can be generated via a cryptic splicing event in the 3′UTR, as indicated by putative splicing signals (GU-AG with an inner distance greater than 60 nt) or by RNA-seq-supported exon junctions in the 3′UTR, and via nonproductive splicing patterns under UPF1-depleted conditions (see "Filtering genes with nonproductive splicing forms" in the Methods section for more details). Because dEJs might be in unannotated or unassembled transcripts, particularly in NMD-responding RNAs, all potential dEJs in all-annotated and newly assembled transcripts were further examined by identifying predicted exon junctions or RNA-seq-based exon-junction signals in control and UPF1-depleted cells. The NMD classifier[52] detected EJC-dependent NMD transcripts with exon inclusion/exclusion or changes in exon start/ends that produced dEJs. Thus, any gene with at least one EJC-dependent NMD-sensitive transcript from a nonproductive splicing form was filtered out. To reduce the random variability in expression fold-changes, genes with fewer than three fragments per million mapped reads in the RNA-seq data of UPF1-knockdown cells were excluded, and to avoid the divide-by-zero error, those showing no expression in control cells were excluded. In the next step, the putative off-targets of siRNAs against UPF1 and GFP (for ref. [31,35]) and scrambled siRNA (our experiment) were filtered out (see "Filtering siRNA off-targets" in the Material and Methods section for more details).

**Filtering genes with nonproductive splicing forms that generate dEJs**. To identify genes with nonproductive splicing forms that generate dEJs, we performed transcriptome assembly using RNA-seq data from UPF1-depleted cells and applied the resulting transcriptome data to an NMD classifier[52]. For transcriptome assembly, RNA-seq data from UPF1-depleted HeLa cells were mapped to the genomes (hg19: GRCh37.66 for humans; mm9 for mice) using TopHat (version 2.0.6). The mapped reads were subjected to Cufflinks (version 2.1.1) for reference-based transcriptome assembly (parameters: --GTF-guide,–min-intron-length 61, and --max-intron-length 265,006).

**Filtering siRNA off-targets**. Because siRNA off-targeting is mediated by a partial complementarity between siRNA and 3′UTRs, similar to miRNA targeting, the same criteria used for miRNA 7-mer target sites were used to detect siRNA off-targets. mRNAs with 8-mer, 7m8, or 7A1 site(s) of an siRNA in the 3′UTR were regarded as putative siRNA off-targets. Although not all putative off-targets are true off-targets, these mRNAs were rigorously filtered out.

**Preprocessing of high-throughput sequencing data**. Our RNA-seq data were trimmed by 9 and 6 nt for the first and second fragments of paired-end reads, respectively, to remove the adaptor sequence using seqtk version 1.0. All RNA-seq reads were trimmed with a minimum quality of 20 and reads less than 20 nt in length were removed using Sickle version 1.2 (parameters: -q 20 -l 20). Reads were mapped to the reference genomes (hg19 for human and mm9 for mouse) using TopHat version 2.0.6[53], allowing at most five genomic loci mapping, two nucleotide mismatches, and novel introns within 1–99th percentile length of known introns, but not allowing any mismatch at splice sites (parameters: solexa1.3-quals, splice-mismatches 0, min-intron-length 61, max-intron-length 265006, max-multi hits 5, read-mismatches 2). Expression levels and fragments per kilobase of transcript per million mapped reads (FPKM) were calculated by Cufflinks version 2.1.1[54]. For mRNAs with multiple isoforms, we chose the isoform with the longest 3′UTR to measure its expression. Raw CLIP-seq data of HeLa cells were mapped to the human reference genome (hg19) using Bowtie version 4.4.7[55], allowing two mismatches but not allowing multi-loci mapping (parameters: v 2, m 1, best, strata). All replicates were combined using Bed tools version 2.17.0[56]. To identify Ago2- or UPF1-binding sites in the 3′UTR of mRNAs, the findPeaks program of HOMER package[57] was used. For mES cells, Ago2- and UPF1-binding sites processed from CLIP-seq data were analyzed.

**Cumulative distribution function (CDF) analysis**. For both RNA-seq and microarray data, mRNA expression $\log_2$ fold-changes between siRNA-treated and control cells were normalized to the median $\log_2$ fold-change value of mRNAs within the shortest 3′UTR bin (50–349 nt). The distribution of the values was transformed to a CDF to evaluate the statistical significance of the difference

among groups from different 3′UTR bins using R function, ecdf (version 3.1.2). Statistical significance was tested by the K–S test.

**7-mer enrichment analysis**. All possible 7-mers, including CCUG[AG][AG][AG] and miRNA 7-mer sites (7-mer-m8 (7m8), and 7-mer-A1(7A1)), were extracted from the 3′UTR of UMD targets, and the count of each 7-mer was compared to that of the 7-mer extracted from number-matched dinucleotide shuffled sequences of the 3′UTRs. The $P$ value for 7-mer enrichment was estimated by Fisher's exact test. miRNA 7-mer sites, significantly enriched in UPF1-knocked down cells ($P < 0.005$), were considered significant sites. 7-mers that were more significant than the most significant miRNA 7-mer site (miR-16-5p) were subjected to sequence logo analysis.

**Signal-to-noise ratio of triplet nucleotides (3-mers)**. All possible triplet nucleotides were extracted from the 3′UTR of UMD targets and from sequences reverse complementary to miRNAs, and the observed count of each triplet nucleotide was compared to that (expected) of the triplet nucleotides extracted from number-matched dinucleotide shuffled sequences. The ratio of observed versus expected counts was regarded as the signal-to-noise ratio of each triplet nucleotide.

**miRNA target sites**. The expression values of miRNAs were downloaded from previous studies in HeLa[37] and mES[58] cells. For the 10, 30, and 50 most abundant miRNA families in HeLa and mES cells, 8-mer (seed-pairing + base-pairing at 8th position + A at the position opposite to the 1st nucleotide of miRNA), 7m8 (seed-pairing + match at 8th position), 7A1 (seed-pairing + A at the position opposite to the 1st nucleotide of miRNA), 6-mer (seed-pairing), and offset-6-mer (base-pairing at 3 to 8th nucleotides of miRNA) sites were considered miRNA target sites. Of these sites, 7-mer sites, including 8-mer, 7m8, and 7A1 sites, were analyzed as effective target sites. For each mRNA, effective target sites were counted in the 3′ UTR and updated with a major 3′UTR end, and the number of sites was correlated to the 3′UTR length and mean expression fold-change in UPF1-knockdown cells.

**Generation of random controls and statistical tests**. To build random controls of dEJ-free genes, 275 nontarget mRNAs lacking any 6-mer site for the 50 most abundant miRNAs were selected. Because the target mRNAs with 7-mer sites generally had longer 3′UTRs than the nontarget mRNAs with no target sites, control of the 3′UTR length is required for miRNA target expression analysis. To compare the changes in expression of the miRNA targets with those of the nontargets, nontargets that embedded 7-mer sites of number-matched random miRNAs were selected as random negative controls. The random miRNA sequences were generated by dinucleotide-shuffling of the 50 most abundant miRNAs using the Python random module. To test the statistical significance, 1000 random miRNA cohorts (each included 50 random miRNA sequences) and 1000 corresponding random control cohorts (each included random targets that embedded 7-mer sites of the random miRNAs) were generated. For one sample $t$ test, the mean values of the random control cohorts (that followed the normal distribution based on the central limit theorem) were compared to those of the targets to measure the $P$ values.

**Expression changes of miRNA targets with effective 7-mer sites**. The mean changes in the expression of targets with effective 7-mer sites of miRNAs in UPF1-, Dicer1-, SMG7-, and TNRC6-knockdown cells were normalized to those of random controls. The mean changes in expression were sorted by the number of sites and wContext + + score.

**wContext + + score calculation**. All wContext + + scores were downloaded from TargetScan v7.1 Human and Mouse[41]. For each target, all wContext + + scores for 7-mer sites of miRNAs were summed. If sites of different miRNAs overlapped in the 3′UTR, the site showing greater efficacy was considered.

**Cell culture**. HeLa cells (KCLB #1002) were maintained in DMEM supplemented with 10% fetal bovine serum and 1% penicillin/streptomycin, and cells were tested for mycoplasma. No additional cell authentication was performed. Cells were seeded one day before transfection with plasmid DNA, siRNA, or miRNA inhibitor/mimic using Lipofectamine 3000 (Invitrogen, Carlsbad, CA, USA). After incubation, the cells were lysed with hypotonic buffer for western blotting or IP, as previously described[59,60], or TRIzol reagent (Invitrogen) for RT-qPCR or RNA-seq. When indicated, FLAG-tagged Ago2[61], MYC-tagged UPF1 WT, or the helicase mutant R843C[62] was transfected into HeLa cells. To silence the expression of specific endogenous genes, the specific siRNAs or antisense oligos (ASOs) chemically modified with phosphorothioate in the backbone and constrained methyl modifications at the 5' and 3' ends. All the sequences of siRNAs and ASOs are listed in Supplementary Table 7. As a control, cells were transfected with nonspecific siControl for siRNA-mediated downregulation or a random ASO for ASO-mediated downregulation (Integrated DNA Technologies, Inc., Coralville, IA, USA). The specific miRNA inhibitor and pseudo-mimic were purchased from Integrated DNA Technologies, Inc.

**IP and western blotting**. To examine whether FLAG-Ago2 forms a complex with UPF1 and SMG7, HeLa cells that were transiently transfected with FLAG-Ago2 were employed for IP using FLAG antibody-conjugated beads (Sigma, St. Louis, MO, USA) as previously reported[59,63]. Co-immunoprecipitated proteins were analyzed by western blotting. The proteins in cell lysates and IP eluates were separated by electrophoresis on sodium dodecyl sulfate polyacrylamide gels and transferred to nitrocellulose membranes (GE Healthcare Life Sciences, Little Chalfont, UK). The specific proteins were probed with antibodies against the following proteins: FLAG (GenScript, Piscataway, NJ, USA, A00187-100, 1:1000), UPF1 (Cell Signaling Technology, Danvers, MA, USA, #9435, 1:1000), Dicer (Cell Signaling Technology, #5362, 1:1000), SMG7 (Bethyl Laboratories, Montgomery, TX, USA, A302-170A, 1:1000), β-actin (Sigma, A2228, 1:2000), PABP (Santa Cruz Biotechnology Dallas, TX, USA, sc-28834, 1:500), Ago2 (Abnova, Taipei, Taiwan, H00027161-M01, 1:1000), NOT1 (Proteintech, Rosemont, IL, USA, 14276-1-AP, 1:500), NOT3 (Abcam, Cambridge, Cambridgeshire, UK, ab154276, 1:500), TNRC6A (Abcam, Cambridge, Cambridgeshire, UK, ab156173, 1:500) TNRC6C (Bethyl Laboratories, Montgomery, TX, USA, A303-969A, 1:500) and Calnexin (Santa Cruz Biotechnology, Dallas, TX, USA, sc-11397, 1:1000). Uncropped western blotting images are provided in the Supplementary Information.

**RT-qPCR**. Total RNA was extracted using TRIzol (Invitrogen) according to the manufacturer's protocol. Next, RNA was further treated with RQ DNase I (Promega, Madison, WI, USA) for 30 min at 37 ℃ to eliminate possible exogenous or endogenous DNA contamination. The RNA was reverse-transcribed using random hexamers (Macrogen, Seoul, Korea), and reverse-transcribed cDNA was amplified by RT-qPCR. PCR was performed using the primers listed in Supplementary Table 8.

**Quantification of miRNA**. miRNAs were quantified using QuantiMir (Systems Biosciences, Palo Alto, CA, USA) according to the manufacturer's protocol. Briefly, a poly(A) tail was added to purified total RNA using poly(A) polymerase and OligodT adaptor. RT-qPCR was performed using the primers listed in Supplementary Table 9.

**Plasmid construction**. To construct pmirGLO-PEA15-WT and pmirGLO-PEA15-Mut-A and pmirGLO-PEA15-Mut-B, the pmirGLO plasmid (Promega) was digested with XhoI and XbaI, and the digested vector fragment was ligated to a PCR product digested with the same enzymes. The PCR product was obtained by annealing three primers named PEA15-5′, PEA15-middle-WT or PEA15-middle-Mut-A or PEA15-middle-Mut-B, and PEA15-3′, where PEA15-middle-Mut contained mutated sequences, as indicated in Fig. 5e; the sequences were amplified using two primers named PEA15-XhoI-F and PEA15-XbaI-R. Similarly, pmirGLO-RPS19BP1-1 and pmirGLORPS19BP1-2 were constructed. Similarly, the miR-24-3p-binding sites in the 3′UTRs of the DNAL4, ATAD2B, ELL, PAK4, DBNDD1, and ZNF740 genes were cloned into reporter constructs. All primer sequences are provided in Supplementary Table 10.

To construct pFLAG-SMG7-WT or mutant (ΔPC) plasmids, which cannot bind to the deadenylase complex[28], the pFLAG plasmid (Sigma) was digested with ClaI and KpnI, and the digested vector fragment was ligated to a PCR product digested with the same enzyme. The PCR product was obtained using pEYFPN1-hSMG7 as a template[64] and two primers, SMG7-WT-1-ClaI-F and SMG7-WT-3276-KpnI-R (for WT) and SMG7 Mut-1876-KpnI-R (for a mutant, ΔPC).

**Reporting summary**. Further information on research design is available in the Nature Research Reporting Summary linked to this article.

## Data availability

Raw RNA-seq data have been deposited in the NCBI Gene Expression Omnibus (GEO; https://www.ncbi.nlm.nih.gov/geo/) under accession number GSE99169. All data are available from the authors upon reasonable request.

## Code availability

All python and R codes used in this manuscript are available on GitHub repository (https://github.com/jwnam/UPF1).

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

## Acknowledgements

We thank Dr. Narry Kim (Seoul National University) for the pCK-FLAG-Ago2 plasmid. This work was supported by National Research Foundation of Korea (NRF) grants funded by the Korean Government (2017M3A9C8028794 to J.H., 2017M3A9G8084539 to J.N., 2018R1A2B2003782 to J.N.) and by a grant from the Medical Research Center (2017R1A5A2015395 to J.H.) funded by the NRF of Korea of the Ministry of Science and ICT, Republic of Korea.

## Author contributions

J.W.H. and J.-W.N. conceived and devised the study. J.Y.P and N.R.A. performed all the experiments. J.-W.S. and S.J.P. performed the bioinformatics and statistical analysis. J.W.H. and J.-W.N. supervised the research and wrote the paper together with J.-W.S.

## Additional information

**Competing interests:** The authors declare no competing interests.

