## [Peer Review File · Nature Communications]

Reviewers' comments:

Reviewer #1 (Remarks to the Author):

Ibrahim and colleagues report in this manuscript the involvement of the NMD factors UPF1 and SMG7 in a TNRC6-independent form of miRNA-mediated downregulation of mRNAs possessing one or several binding sites for the respective miRNAs in their 3' UTRs.

From different published and their own mRNA-seq data, the authors have selected genes that do not express any mRNA isoform annotated to have the stop codon >50 nts upstream of the last exon-exon junction and analyzed these so-called non-NMD targets for increased mRNA abundance in response to UPF1 knockdown (kd). As expected from previous work, a substantial fraction of these transcript increased in abundance upon UPF1 kd and among these, transcripts with long 3' UTRs were enriched.

In the absence of Dicer, and hence reduced miRNA levels, this UPF1-dependent de-repression of the "non-NMD targets" was lost, leading the authors to propose that this de-repression was dependent on the miRNAs predicted to bind to these mRNAs. Indeed, the 3'UTR length-dependent upregulation of mRNAs upon UPF1 kd correlated with the number of predicted miRNA binding sites in a given 3' UTR and this correlation was lost in the background of Dicer kd. Fig. 4 shows that among nine selected genes with long 3' UTRs and binding sites for miR-24-3p, the de-repression observed upon miRNA inhibition was very small (< 1.75 fold) while the de-repression upon UPF1 kd was still moderate but somewhat stronger (< 5 fold). While clearly both UPF1 and miR-24-3p appear to downregulate most of these transcripts, it remains unclear if the two effects are mechanistically linked or completely independent of each other. To claim from the presented data that UPF1 and miRNA function in a cooperative manner (lines 271-72) is an over-interpretation, as "cooperativity" implies a mechanistic link. The term additive effect would be more appropriate. Generally, most of the results are displayed in a double- or even triple-normalized way and the effects are often quite small. This is problematic, because it is not possible for the reader to judge if the reported effects indeed mainly result from changes of the tested mRNAs with all normalizers remaining very constant or if a combination of minimal changes in the tested mRNAs together with minimal variations in the normalizers produce the presented results. Whenever possible, less processed data should be shown, which is also easier to comprehend and interpret. Furthermore, it was not clear to me if the authors attempted a transcript specific analysis of the short-read Illumina data based on annotated splice forms or if their analysis was done at gene level. Both analysis pipelines have their caveats. Isoform-specific assignments of reads based on known annotations is not working well when studying NMD, because non-productive splice forms producing PTCs are often not annotated in the databases and hence will be missed. Similarly, if the analysis was limited to gene level, it is possible that the de-repression observed with kd of NMD factors is taking place on different transcript isoforms than the de-repression observed by inhibiting miRNA-mediated mRNA turnover.

The evidence to claim a mechanistic involvement of UPF1 and SMG7 in miRNA-mediated mRNA turnover is weak and essentially based on the co-IP of UPF1 and SMG7 with FLAG-AGO2.

Although the authors state that this interaction was not RNA-mediated, the data shown in Fig. 7A shows a significant decrease in co-precipitated UPF1 and SMG7 when the lysate was treated with RNase A before IP. Furthermore, the distinction between a TNRC6-dependent miRNA-mediated mRNA downregulation and a UPF1-SMG7-dependent but TNRC6-independent pathway is not compelling and mainly built on negative results (i.e. no change in TNRC6 kd). Negative results of kd experiments are always inconclusive, since it cannot be excluded that the kd was not strong enough. Collectively, the presented data is suggestive for a mechanistic link between the NMD factors UPF1-SMG7 and AGO2-miRNA mRNA regulation but not compelling.

Additional points:

- Lines 59 – 61: There seems to be a mix up of references. The wrong papers are cited for the respective three distinct decay pathways.
- Line 98: The Results section would benefit from a few introductory sentences describing the rationales and aims of the chosen approach.

- Line 117: Please clarify: was the cut off for de-repression $\log_2(0.2) = -2.322$. For most readers, it would be easier to understand if the greater than 2.3 fold de-repression was indicated rather than the corresponding \log_2 value. It would also be prudent to indicate the number of transcripts that decreased in abundance by ≥ 2.3 fold upon UPF1 kd (one would expect much fewer than those increasing).
- Table S1: Define UMD
- Line 175: Correct syntax: similarly to in conserved
- Supplementary Fig. 4A,B: Please indicate if this comparison is filtered for CLIP sites exclusively in 3' UTRs or if this compares all UPF1 CLIP sites with all AGO2 CLIP sites.
- Some of differences indicated as statistically significant are very small. Please give information on the number of independent biological replicate experiments that were measured and averaged here. From our own experience with RT-qPCR experiments, changes of less than 2-fold are hardly significant if based on the typical 3 independent biological replicates.
- Lines 356-358: This is an incomprehensible sentence. Please reformulate and perhaps split the information in two separate sentences.
- Fig. 7: What is the reason to show Calnexin in the input, when it is then not probed for in the IPed samples? I suggest to delete it.
- Fig. 7C-F: The results are inconclusive because there is no evidence that the TNRC6A/C kd had an effect at all. Examples of miRNA-mediated TNRC6-dependent mRNA decay should be included as positive controls.
- Lines 465-468: Since you compare knockdowns and use expression change thresholds, the partial overlap between the UPF1-dependent and the SMG7-dependent targets could simply be technical and does not necessarily indicate additional pathways involving other factors than SMG7.
- The model shown in Fig. 7H proposes that UPF1 bound to a CUG sequence will be displaced from its binding site by binding of the AGO2/miRNA complex yet remain in the complex bound to AGO2. There is no experimental evidence for this rather unlikely scenario.

Reviewer #2 (Remarks to the Author):

The manuscript dissects the interaction between the NMD machinery, specifically UPF1 and SMG7 proteins, and the miRNA pathway, describing a potentially novel mechanism of miRNA mediated mRNA decay. The authors show that UPF1 regulates the stability of mRNAs by a mechanism independent of NMD, but dependent on the 3'-UTR length. The 3'UTR of the mRNAs undergoing UPF1-dependent and NMD-independent mRNA decay (UMD) are enriched for miRNA recognition elements and the CUG motif. Given that in UPF1/Dicer1 double knockdown cells the regulation of these mRNAs and their 3'UTR dependence is lost, the authors conclude that UMD requires miRNA mediated regulation. It is shown that UPF1-dependent miRNA targeting is in part mediated by the UPF1 – SMG7 interaction. Moreover, it is shown that Ago2, UPF1 and SMG7 interact, and that the SMG7 interaction with Ago2 is UPF1-mediated. However, the authors indicate that UPF1-dependent miRNA target decay is TNRC6-independent. Finally, in the proposed model of an alternative miRNA targeting pathway the Ago2-UPF1 interaction recruits SMG7, which would mediate the recruitment of CCR4-NOT deadenylation complex.

Although this is a potentially interesting manuscript, there are some technical issues and inconsistencies that should be thoroughly addressed.

1. In the Results section, the authors state that they have 'removed all siRNA off-targets through our computational pipeline' (page 6, lines 108-110). How did the authors determine which are the siRNA off-targets? Additional details have to be provided considering this matter and the approach used to filter the off-target effects.
2. The authors state that the bin sizes for 3'UTR lengths were 'heuristically set with more data in the first and last bins' (page 7, line 119). The bin size choice should be reevaluated and the

strategy for choice of the 3'UTR lengths bins thoroughly explained.

3. Why do the authors in Figure 3C show the CDF graph for expression changes in Dicer1/UPF1 knockdown cells compared to Dicer1 siRNA? It would be more interesting to compare to UPF1 siRNA treated cells. Moreover, the CDF graph for expression changes in Dicer1 compared to control siRNA should be shown.

4. In page 12 (lines 233-235) the authors state that 'As expected, expression changes in targets with an Ago2-UPF1 overlapping site were slightly more de-repressed than those targets with a single Ago2 site'. First, the statement should be modified given that there is no statistical difference ($P=0.07$). Second, why do the authors expect that Ago2-UPF1 overlapping sites would be more derepressed than with Ago2 alone?

5. In Figures 4, 6 and 7 the authors are using 9 putative miR-24-3p targets. However, it is not shown in the manuscript that these are indeed miR-24-3p direct targets. This should be performed to substantiate the interpretation of the results shown in Figures 4, 6 and 7.

6. Along the same line, miRNA inhibitors against miR-24-3p and miR-26a are used (Figures 4 and Supplementary S4), however the significance of the inhibitor effect on the putative target expression is not shown. For most cases it appears to be a minor effect.

7. Clarify the use of 'pseudo-mimic strategy' instead of classical miRNA mimics. This is particularly important since it has been thoroughly shown that the base pairing outside of the seed-sequence is important for target regulation.

8. Indeed, the experiments using miRNA mimics and inhibitors are shown in rather complicated ways. An extreme example are the results shown in Figure 4F using a double normalization, resulting in difficult evaluation and interpretation of the results.

9. The efficiency of TNRC6A/C knockdown should be demonstrated by Western-blotting.

10. Along this line, positive controls for TNRC6A/C knockdown in terms of target repression should be shown using canonical miRNA targets (Ago2/TNRC6-dependent, UPF1-independent).

11. It would be most interesting and significantly strengthen the manuscript to demonstrate that, in the context of UMD, SMG7 is indeed recruiting the deadenylase complex to mediate miRNA-mediated decay.

12. Clarify the reason of the use of the term 'endogenous miRNAs' throughout the manuscript.

Reviewer #3 (Remarks to the Author):

This study indicates that miRNAs containing CAG triplets within seeds may regulate targets through an alternative miRNA-mediated repressive pathway, which is dependent from UPF1/SMG7 but independent from TNRC6.

In general, I believe that this is a highly important claim, but it is not yet well supported experimentally and bioinformatically. I am a bit puzzled regarding the control transcript sets used during the bioinformatic analyses and this should be better defined and clarified. Also, this study would significantly benefit from the use of CRISPR-CAS9 KO lines, to conclusively exclude siRNA-off target issues.

Specifically:

Page 6, lines 98 to 103. This part is unclear, the authors should explain it better. Which databases were used exactly here? It lacks of a small introduction and therefore it is arduous to understand. Difficult to comment on this part

It is unclear why replication-dependent histone genes should be excluded from the analysis, can this be clarified?

The authors should explain in details the strategy used to remove off-targets of siRNAs against UPF1, GFP and scrambled siRNAs. They refer to their "computational pipeline" that I could not find anywhere nor in the results section neither in the material and methods.

In addition, I do not believe that removing off target bioinformatically is the correct step here because this procedure is based on prediction only. I would suggest to the authors to additionally perform their own RNA-seq experiments from CRISPR-UPF1-KO and control lines, also.

It is unclear why the authors choose a cut-off of log₂ fold-change = 0.2 to select UPF1 targets. This filter looks too relaxed to me and could include a large number of off-targets.

The authors should also perform RNA-seq experiments using double KO or Kd lines for UPF1 and another component of the canonical NMD pathway (i.e. SMG6) to conclusively demonstrate their hypotheses.

Figure 2a. The authors highlight highest ranked 7-mers in grey which show a specific motif predominately starting with C or T, as evidenced by the motif logo on top of the graph. It is not clear from the manuscript the significance of this aspect.

Line 187-193, Figure 3a. It is unclear which transcripts are used as control here and the number of transcripts involved in the control signature. How could the authors select transcripts that lack endogenous 7-mer sites but contain random 7-mer sites? Transcripts lacking 7-mer sites are theoretically rare. Is the number of these transcripts enough for robust statistical testing? Also, why this analysis was not performed using cumulative distribution function?

The RT-qPCR experiment in Supplemental Figure 3c lacks statistical test because probably performed only once. In addition, the one in Supplemental Figure 3d is not convincing because log₂ fold-change is too low. This cannot conclusively demonstrate that Dicer1 was kept downregulated for long enough to give the possibility to miRNAs, which tend to be particularly stable, to be down-regulated during this experiment. I would suggest the use of CRISPR-CAS9 Dicer1 KO lines, here.

Line 196. Also this experiment should be repeated with CRISPR-CAS9 lines.

In lines 244 to 248 the authors try to select UPF1-dependent miR-24-3p targets considering transcripts with miR-24-3p 8-mer or 7m8 mers, but no sites for other miRNAs. I looked at those transcripts in TargetScan and they actually appear to contain plenty of sites for other miRNAs, which are also conserved across species and repeated multiple times within the 3'UTRs (see for example miR-29-3p for ATAD2B, miRNA that does not contain CAG within its seed) indicative of active repression. The author should demonstrate that these transcripts are exclusively targeted by miR-24-3p, but not other miRNAs to consider this gene signature appropriate for following analyses. Same thing is valid for miR-26a

To conclusively demonstrate that TNRC6 is not involved in this miRNA-repressive pathway the authors should perform RNA-seq from TNRC6 CRISPR-CAS9 stable KO cellular clones.

Minor points

The authors should remove endo-miRNAs throughout the manuscript and replace this term with miRNAs, because contrarily to siRNAs, miRNAs are endogenous by definition.

We thank all three reviewers for their many helpful comments and insights, which have led to substantial improvements to our paper. We have performed most of the experiments and computational analyses suggested by the reviewers and have revised the manuscript to address the concerns expressed by the reviewers.

Furthermore, (1) we have reperfomed the computational analyses and reproduced all the results with PTC-free mRNAs after filtering genes with nonproductive splicing isoforms embedding PTC. (2) To strengthen the evidence for the functional link between AGO2-mediated miRNA targeting and the UPF1-SMG7-NOT1/3 complex, we observed the coimmunoprecipitated deadenylase complex in the presence or absence of UPF1 and tested the mechanistic model using a SMG7 deletion mutant that could not bind to the deadenylase complex.

We hope that we have adequately responded to the inquires of the reviewers and that you will find our revised manuscript suitable for publication.

Reviewers' comments:

Reviewer #1 (Remarks to the Author):

Ibrahim and colleagues report in this manuscript the involvement of the NMD factors UPF1 and SMG7 in a TNRC6-independent form of miRNA-mediated downregulation of mRNAs possessing one or several binding sites for the respective miRNAs in their 3' UTRs. From different published and their own mRNA-seq data, the authors have selected genes that do not express any mRNA isoform annotated to have the stop codon >50 nts upstream of the last exon-exon junction and analyzed these so-called non-NMD targets for increased mRNA abundance in response to UPF1 knockdown (kd). As expected from previous work, a substantial fraction of these transcript increased in abundance upon UPF1 kd and among these, transcripts with long 3' UTRs were enriched. In the absence of Dicer, and hence reduced miRNA levels, this UPF1-dependent de-repression of the "non-NMD targets" was lost, leading the authors to propose that this de-repression was dependent on the miRNAs predicted to bind to these mRNAs. Indeed, the 3'UTR length-dependent upregulation of mRNAs upon UPF1 kd correlated with the number of predicted miRNA binding sites in a given 3' UTR and this correlation was lost in the background of Dicer kd.

1) Fig. 4 shows that among nine selected genes with long 3' UTRs and binding sites for miR-24-3p, the de-repression observed upon miRNA inhibition was very small (< 1.75 fold) while the de-repression upon UPF1 kd was still moderate but somewhat stronger (< 5 fold). While clearly both UPF1 and miR-24-3p appear to downregulate most of these transcripts, it remains unclear if the two effects are mechanistically linked or completely independent of each other. To claim from the presented data that UPF1 and miRNA function in a cooperative manner (lines 271-72) is an over-interpretation, as "cooperativity" implies a mechanistic link. The term additive effect would be more appropriate.

(Response) Thank you for the valuable comment. As you mention, UPF1 KD exhibits stronger effects than inhibition of miR-24-3p (Fig. 4b and e). Because the nine selected genes could be targeted by other miRNAs, which is partly related to the UMD, UPF1 KD could have greater effects on targets than the inhibition of a single miRNA. Fig. 4e and 4f demonstrate the interaction between miRNA and UPF1-mediated gene regulation by analyzing the effects of siUPF1 in the presence and absence of miR-24-3p. However, because these results do not explain whether the interactions are mechanistically linked, we have changed "cooperative..." to "additive..." to clarify our intended meaning, as suggested by the reviewer.

2) Generally, most of the results are displayed in a double- or even triple-normalized way and the effects are often quite small. This is problematic, because it is not possible for the reader to judge if the reported effects indeed mainly result from chances of the tested mRNAs with all normalizers remaining very constant or if a combination of minimal changes in the tested mRNAs together with minimal variations in the normalizers produce the presented results. Whenever possible, less processed data should be shown, which is also easier to comprehend and interpret.

(Response) As suggested by the reveiwer, some bioinformatic results (Fig. 3a, 3d, and 6d; Supplementary

Fig. 3d, 6g, 7c and 7d) were updated with less normalized or prenormalized values in the revised manuscript. For instance, the current Fig. 6d displays the preprocessed results, including the changes in expression in the random control, as shown below.

The revised Fig. 6d (left) and the original figure (right)

In addition, two experimental results (Fig. 4f and Fig. S4i) with triple normalization were reproduced with a lower degree of normalization. The less processed results in Fig. 4f and S4i (in the original and current manuscript) have been provided in the “Reviewer only version” as shown below. Briefly, the levels of the miRNA targets increased upon UPF1 depletion and treatment with a miRNA inhibitor, which reduced the amount of miRNA target transcripts, as observed in Fig. 4b and 4d. Downregulation of UPF1 and treatment with a miRNA inhibitor increased the miRNA target transcript levels. To determine the effects of UPF1 on the miRNA targets, we tried to compare the effects of UPF1 depletion in the presence or absence of a miRNA inhibitor. However, the unprocessed results shown below are more difficult to interpret than intended. Thus, we determined the ratios of the transcript levels under UPF1 depletion in the presence or absence of the miRNA inhibitor, as shown in the original Fig. 4f, demonstrating that the effects of UPF1 depletion were masked by the miRNA inhibitor. While we agree with the reviewer’s comment, we believe that the present, more processed, results will be more useful to the readers. If the reviewer suggests that we include the unprocessed results in the manuscript, we will include the “Reviewer only version” shown below in the manuscript.

Corresponding to Fig. 4f

Corresponding to Fig. S4i

Many double normalizations, included in Fig. 4b, 4d, 4e, 5f, 5g, 5h, 5i, 6h, 7d and 7e, could be converted to single normalization in the figures, but we believe that these results will be very distracting because we verified as many transcripts as possible. Furthermore, we included the target transcripts but also several negative and positive controls, as previously reported (see Reviewer #2, question 6), and performed independent biological experiments enough times to attain statistical significance (we have included the number of experimental repeats in the revised manuscript). Thus, we strongly believe that our quantification is reliable, although the variations are not large. We hope the reviewer accepts our explanation.

3) Furthermore, it was not clear to me if the authors attempted a transcript specific analysis of the short-read Illumina data based on annotated splice forms or if their analysis was done at gene level. Both analysis pipelines have their caveats. Isoform-specific assignments of reads based on known annotations is not working well when studying NMD, because non-productive splice forms producing PTCs are often not annotated in the databases and hence will be missed. Similarly, if the analysis was limited to gene level, it is possible that the de-repression observed with kd of NMD factors is taking place on different transcript isoforms than the de-repression observed by inhibiting miRNA-mediated mRNA turnover.

(Response) Thank you for the valuable comment. We processed all the reads to estimate expression at the gene level. To identify unannotated, potential NMD target genes in addition to known NMD targets, we examined whether these genes had nonproductive, alternatively spliced transcripts that could produce PTCs in siUPF1-treated cells. For this purpose, we sought to reconstruct transcriptomes using reference-based transcriptome assembly and identified transcripts that generate PTCs by nonproductive splicing. In addition, we excluded any gene with cryptic splicing signals in the 3'UTR to ensure the removal of potential NMDs. Thus, if a gene includes any transcript with a PTC by nonproductive splicing, the locus was excluded without considering the changes in expression in our analysis. Once we excluded all known and potential NMD target genes, the changes in expression of the PTC-free genes were analyzed at the gene level to assess the response of the UPF1 knockdown. To clarify our intended meaning, we changed the term "PTC-free mRNAs" to "PTC-free genes" throughout the manuscript, unless the indicated molecules were indeed RNAs, and added a detailed description of the method in the sections "PTC-free genes" and "Filtering genes with nonproductive splicing forms that generate PTCs" in the Materials and Methods of the revised manuscript.

4) The evidence to claim a mechanistic involvement of UPF1 and SMG7 in miRNA-mediated mRNA turnover is weak and essentially based on the co-IP of UPF1 and SMG7 with FLAG-AGO2. Although the authors state that this interaction was not RNA-mediated, the data shown in Fig. 7A shows a significant decrease in co-precipitated UPF1 and SMG7 when the lysate was treated with RNase A before IP. Furthermore, the distinction between a TNRC6-dependent miRNA-mediated mRNA downregulation and a UPF1-SMG7-dependent but TNRC6-independent pathway is not compelling and mainly built on negative results (i.e. no change in TNRC6 kd). Negative results of kd experiments are always inconclusive, since it cannot be excluded that the kd was not strong enough. Collectively, the presented data is suggestive for a mechanistic link between the NMD factors UPF1-SMG7 and AGO2-miRNA mRNA regulation but not compelling.

(Response) Thank you for the valuable comments. In addition to the co-IP of UPF1 and SMG7 with FLAG-AGO2, the AGO-UPF1-SMG7 axis is supported by the results that the siSMG7 conditions phenocopy those of siUPF1 in terms of miRNA-dependent UPF1-mediated regulation in the original study (Fig. 6). To determine whether FLAG-AGO2 could interact with the deadenylase complex via the UPF1-SMG7 complex, NOT1 and NOT3, which are components of the deadenylase complex, were immunoprecipitated using an anti-FLAG-antibody in the presence or absence of UPF1. We found that SMG7-NOT1/NOT3 physically interacted with AGO2 in a UPF1-dependent manner and have described the results in Fig. 7b (in the revised manuscript). We also agreed with the reviewer's point that UPF1 and SMG7 interact with FLAG-AGO2 in a partially RNA-independent manner, which has been included in the revised manuscript.

Revised Fig. 7b.

To further confirm the mechanistic link between UPF1-SMG7 and AGO2-miRNA-mediated mRNA regulation, we constructed a SMG7 deletion mutant (Δ PC) that could not bind to the deadenylase complex¹ and tested our mechanistic model (AGO2-UPF1-SMG7-deadenylase complex). Briefly, we overexpressed wild-type FLAG-tagged SMG7 and the FLAG-tagged SMG7 mutant (Δ PC) and quantified the UMD targets listed in Fig. 4 and 7. The RT-qPCR results indicated that most of the UPF1/SMG7-dependent transcripts, except *ATAD2B*, were upregulated by overexpression of FLAG-SMG7 Δ PC compared to the levels observed upon overexpression of FLAG-SMG7 WT (Supplementary Fig. 7b in the revised manuscript), suggesting that the loss in the SMG7-deadenylase complex increased the levels of the miR-24-3p targets. We hope that the reviewer will find that these new results strongly support our mechanistic model.

Revised Supplementary Fig. 7b.

Additional points:

- Lines 59 – 61: There seems to be a mix up of references. The wrong papers are cited for the respective three distinct decay pathways.

(Response) Thank you for commenting on our mis-citation. We have double-checked the references cited for each pathway and updated the citations. Briefly, Schmidt *et al.* (2015) *Nucleic Acids Research* should have been the article cited for “SMG6-mediated cleavage near the PTC”, and we have corrected this error.

- Line 98: The Results section would benefit from a few introductory sentences describing the rationales and aims of the chosen approach.

(Response) We agree with the reviewer’s suggestion. To describe the rationales and aims of the computational approach for selecting genuine non-NMD genes, we have added a few introductory descriptions and revised the sentences as shown below.

From:

“It is well-known that UPF1 regulates the stability of mRNAs through a long 3’UTR. However, UPF1-dependent targets may include actual dEJs or PTC because the search for dEJs or alternative splicing isoforms was conducted using only a public database rather than by comprehensively analyzing actual dEJs or PTCs. We thus sought to exclude not only known NMD targets, but also all genes with potential PTCs to analyze the expression changes in PTC-free targets in UPF1-depleted cells.”

To:

“Although early studies reported that a considerable fraction of RNAs responsive to UPF1 depletion are dependent on the 3’UTR length, the question of whether UPF1-responsive RNAs actually decay in an NMD-independent manner has been repeatedly asked. To address this unanswered question, we sought to identify genuine non-NMD genes and reanalyze these genes to check whether they continued to exhibit responses in UPF1-depleted cells. To identify non-NMD genes, we first removed all the genes that included any transcript with potential PTCs in the annotation (NCBI RefSeq: see the “Data Source” section of Materials and Methods for additional details) or PTCs in transcripts that were newly assembled from the RNA-seq data of UPF1-depleted cells using our computational pipeline (Fig. 1a; see the “PTC-free genes” and “Filtering genes with nonproductive splicing forms” sections in the Materials and Methods for additional details).”

- Line 117: Please clarify: was the cut off for de-repression $\log_2(0.2) = -2.322$. For most readers, it would be easier to understand if the greater than 2.3 fold de-repression was indicated rather than the corresponding \log_2 value. It would also be prudent to indicate the number of transcripts that decreased in abundance by ≥ 2.3 fold upon UPF1 kd (one would expect much fewer than those increasing).

(Response) Thank you for the comment. We used a cut-off of $\log_2(\text{FC}) = 0.2$ (FC is 1.14 for fold change). This cut-off was chosen because the median effect of transfected miRNAs on targets embedding a 7A1 site was $\log_2(\text{FC}) = 0.2$ in a previous study². In addition, as per the reviewer’s suggestion, we have added the number (1305, 18.3%) of genes with decreased abundance, which was less than $\log_2(\text{FC}) = -0.2$, in the revised manuscript. The revised sentences (page 7 line 132 of the revised manuscript) are shown below.

“Using the RNA-seq data and the pipeline shown in Fig. 1a, we compiled 7,117 PTC-free genes, 3,419 (48.0%) of which were derepressed by a greater than \log_2 -fold change (0.2) upon downregulation of UPF1 (Supplementary Table 1), while 1,305 (18.3%) were downregulated by a less than \log_2 -fold change (-0.2).”

- Table S1: Define UMD

(Response) As per the reviewer’s suggestion, we have added the definition of UMD in Table S1.

- Line 175: Correct syntax: similarly to in conserved

(Response) We have corrected the syntax in the revised manuscript as shown below.

We have changed “*similarly to in conserved*” to “*similarly, CUG was conserved in*”.

- Supplementary Fig. 4A,B: Please indicate if this comparison is filtered for CLIP sites exclusively in 3’ UTRs or if this compares all UPF1 CLIP sites with all AGO2 CLIP sites.

(Response) We compared the Ago2 and UPF1 CLIP-seq sites exclusively in the 3’UTRs. We have included the description in the revised legend of Supplementary Fig. 4a,b.

- Some of differences indicated as statistically significant are very small. Please give information on the number of independent biological replicate experiments that were measured and averaged here. From our own experience with RT-qPCR experiments, changes of less than 2-fold are hardly significant if based on the typical 3 independent biological replicates.

(Response) We have added the number of independent biological experiments used to calculate the *P*-values in all the figure legends. Because the effect of a single miRNA target site ranges from 15% (7A1) to 35% (8 mer)², we needed to test the small changes with a high number of replicates. As per the reviewer's suggestion, we have added the number of replicates in the figure legends of the revised manuscript.

- Lines 356-358: This is an incomprehensible sentence. Please reformulate and perhaps split the information in two separate sentences.

(Response) As suggested by the reviewer, we have simply rephrased the sentence as shown below.

From: "Expression changes in UPF1-dependent miRNA targets embedding significantly enriched 7-mers of Fig. 2a in siSMG7-treated cells explained approximately 40% of the expression changes in siUPF1-treated cells"

To: "In fact, the change in expression of the UMD targets with significant 7-mer sites in siSMG7-treated cells reached approximately 40% of that in siUPF1-treated cells"

- Fig. 7: What is the reason to show Calnexin in the input, when it is then not probed for in the IPed samples? I suggest to delete it.

(Response) As per the reviewer's suggestion, we have excluded the calnexin result in revised Fig. 7.

- Fig. 7C-F: The results are inconclusive because there is no evidence that the TNRC6A/C kd had an effect at all. Examples of miRNA-mediated TNRC6-dependent mRNA decay should be included as positive controls.

(Response) As per the reviewer's suggestion, we performed Western blotting for TNRC6A/C to determine whether TNRC6A/C was efficiently downregulated by siRNA. We now show that TNRC6A/C was efficiently downregulated, as shown in Fig. 7c in the revised manuscript. In addition, as a positive control for the downregulation of TNRC6, we selected several genes, including TNGLN, S1PR2, LOXL2 and STK36 (miR-24-3p targets; UPF independent but TNRC6 dependent), as positive controls, and these genes exhibited derepression in TNRC6-depleted cells but not in UPF1-depleted cells, as expected (Fig. 7d).

Revised Fig. 7c and 7d

- Lines 465-468: Since you compare knockdowns and use expression change thresholds, the partial overlap between the UPF1-dependent and the SMG7-dependent targets could simply be technical and does not necessarily indicate additional pathways involving other factors than SMG7.

(Response) Thank you for the comment. We have revised the description, as shown below.

From: *“Approximately 40% of the mean expression change in endo-miRNA targets (for the top 50 most abundant endo-miRNA families) under UPF1-depleted conditions compared to in the random control was mediated by the UPF1-SMG7 pathway (Fig. 3a, 6d), suggesting that there are additional mediators other than SMG7.”*

To *“Approximately 40% of the mean change in expression of the miRNA targets (for the 50 most abundant miRNA families) under UPF1-depleted conditions, compared to that of the random control, was mediated by the UPF1-SMG7 pathway (Fig. 3a, 6d). This finding could indicate the presence of a mediator other than SMG7 or technical limitations, including limitations in knockdown efficiency or experimental variations.”*

- The model shown in Fig. 7H proposes that UPF1 bound to a CUG sequence will be displaced from its binding site by binding of the AGO2/miRNA complex yet remain in the complex bound to AGO2. There is no experimental evidence for this rather unlikely scenario.

(Response) We agree with the reviewer’s comment. In the revised figure, we have removed the step that describes the binding of UPF1 to a CUG motif after displacement from its binding site by the AGO2/miRNA complex (Fig. 7j).

Reviewer #2 (Remarks to the Author):

The manuscript dissects the interaction between the NMD machinery, specifically UPF1 and SMG7 proteins, and the miRNA pathway, describing a potentially novel mechanism of miRNA mediated mRNA decay. The authors show that UPF1 regulates the stability of mRNAs by a mechanism independent of NMD, but dependent on the 3'-UTR length. The 3'UTR of the mRNAs undergoing UPF1-dependent and NMD-independent mRNA decay (UMD) are enriched for miRNA recognition elements and the CUG motif. Given that in UPF1/Dicer1 double knockdown cells the regulation of these mRNAs and their 3'UTR dependence is lost, the authors conclude that UMD requires miRNA mediated regulation. It is shown that UPF1-dependent miRNA targeting is in part mediated by the UPF1 – SMG7 interaction. Moreover, it is shown that Ago2, UPF1 and SMG7 interact, and that the SMG7 interaction with Ago2 is UPF1-mediated. However, the authors indicate that UPF1-dependent miRNA target decay is TNRC6-independent. Finally, in the proposed model of an alternative miRNA targeting pathway the Ago2-UPF1 interaction recruits SMG7, which would mediate the recruitment of CCR4-NOT deadenylation complex.

Although this is a potentially interesting manuscript, there are some technical issues and inconsistencies that should be thoroughly addressed.

1. In the Results section, the authors state that they have 'removed all siRNA off-targets through our computational pipeline' (page 6, lines 108-110). How did the authors determine which are the siRNA off-targets? Additional details have to be provided considering this matter and the approach used to filter the off-target effects.

(Response) Thank you for the constructive comment. We have removed all siRNA off-targets that include 7-mer sites (miRNA-like target sites: 8mer, 7m8, and 7A1) in the 3'UTRs and have described the method in the "Filtering siRNA off-targets" section in the Materials and Methods. To clarify the description in the main text, we have revised the text as shown below.

From:

"We additionally removed all siRNA off-targets (for UPF1, GFP, and scrambled siRNAs) through our computational pipeline (Fig. 1a; see the Materials and Methods section for more details), ..."

To:

"In addition, we removed all the siRNA off-targets (for UPF1, GFP, and the scrambled siRNAs) that included 7-mer sites in the 3'UTR (see "Filtering siRNA off-targets" section in the Materials and Methods for additional details), ..."

In the "Filtering siRNA off-targets" section in the Materials and Methods, we added the detailed method shown below.

"Because siRNA off-targeting is mediated by a partial complementarity between siRNA and 3'UTRs, similar to miRNA targeting, the same criteria used for miRNA 7-mer target sites were used to detect siRNA off-targets. mRNAs with 8mer, 7m8, or 7A1 site(s) of an siRNA in the 3'UTR were regarded as putative siRNA off-targets. Although not all putative off-targets are true off-targets, these mRNAs were rigorously filtered."

2. The authors state that the bin sizes for 3'UTR lengths were 'heuristically set with more data in the first and last bins' (page 7, line 119). The bin size choice should be reevaluated and the strategy for choice of the 3'UTR lengths bins thoroughly explained.

(Response) As per the reviewer's suggestion, we have reevaluated the results with different bin sizes of 3'UTR length: one that had an equal number of genes in each bin, the same setting as that used by Chris Burge's laboratory³, and the original setting. We found that the results were very similar across different bin settings, as shown below. Originally, we set the bin sizes of 3'UTR lengths with more genes in the first and last bins because the genes in the first and last bins tended to have fewer sites and more off-target sites, respectively, in the 3'UTRs, resulting in fewer PTC-free genes in the bins (Fig. S1a in the original manuscript). However, we have decided to update the results using an equal number of genes in each bin in Fig. 1b-d, Fig. 3c, Fig. 6a, b, Fig. 7f, g, Fig. S1, and Fig. S6b-e in the revised manuscript. To clarify the strategy underlying the choice of bin size, we have updated the description as shown below.

From: "To secure a sufficient amount of data for statistical testing in downstream analysis, the bin sizes according to 3'UTR length were heuristically set with more data in the first and final bins"
 To: "The bin sizes of the 3'UTR lengths were chosen to include an equal number of genes in each bin."
 (page 6, line 124).

From: "As a result, the 3'UTR length dependency on UPF1-dependent expression changes in PTC-free mRNAs remained significant in mES, HeLa, K562 cells and (Fig. 1b, c and Supplementary Fig. 1b,c)."

To: "As a result, the 3'UTR length dependency of UPF1-dependent changes in expression of PTC-free genes remained significant in mES, HeLa, and K562 cells (Fig. 1b,c and Supplementary Fig. 1a,b) and was not dependent on the bin sizes of the 3'UTR lengths (Supplementary Fig. 1c)" (page 7, line 127).

The original Fig. 1b-d shows the CDF graphs with the original bin size settings, as shown below.

Original Fig. 1b-d.

The revised Fig. 1b-d shows the CDF graphs with bin sizes of 3'UTR lengths set to an equal number of genes in all bins, as shown below.

Revised Fig. 1b-d.

The newly added Fig. S1c shows the CDF graphs with bin size settings of 3'UTR lengths the same as those previously used in Hurt et al., as shown below.

Fig. S1c

3. Why do the authors in Figure 3C show the CDF graph for expression changes in Dicer1/UPF1 knockdown cells compared to Dicer1 siRNA? It would be more interesting to compare to UPF1 siRNA treated cells. Moreover, the CDF graph for expression changes in Dicer1 compared to control siRNA should be shown.

(Response) Fig. 3c describes the effect of the introduction of siUPF1 on PTC-free genes in Dicer1-depleted background cells, where miRNAs are depleted. The comparison was used to examine whether UMD targeting is dependent on miRNAs. We noticed that there was a typo in a sentence that could cause confusion; therefore, we have corrected the sentence as follows.

“...compared to those in *Dicer1*-knockdown cells.” instead of “... compared to those in *UPF1*-knocked down cells” (page 10, line 212).

In addition, as suggested by the reviewer, we also compared the changes in expression of PTC-free genes in Dicer1-knockdown cells to those in siControl-treated cells, as shown in the following “Reviewer only” version (Fig. A), and the results suggested that there was 3'UTR-length-dependent derepression of mRNAs in Dicer1-knockdown cells.

Reviewer only version; Fig. A.

Furthermore, the changes in expression of miRNA targets with 7-mer sites in Dicer1-knockdown cells versus siControl-treated cells were also compared to those in random control cells (Revised Supplementary Fig. 3d), and the results suggested that the miRNA targets were derepressed in Dicer1-depleted cells.

Revised Supplementary Fig. 3d.

4. In page 12 (lines 233-235) the authors state that ‘As expected, expression changes in targets with an Ago2-UPF1 overlapping site were slightly more de-repressed than those targets with a single Ago2 site’. First, the statement should be modified given that there is no statistical difference ($P = 0.07$). Second, why do the authors expect that Ago2-UPF1 overlapping sites would be more derepressed than with Ago2 alone?

(Response) Thank you for the valuable comment. We expected that there would be an additive effect, as the Ago2-UPF1 overlapping site could affect target gene expression via both the Ago2-UPF1 and Ago2-TNRC6A/C pathways. However, as noted by the reviewer, not all Ago2-UPF1 overlapping sites could lead

to additive effects; thus, we have revised the description as shown below in the revised manuscript.

From: “As expected, expression changes in the targets with an Ago2-UPF1 overlapping site (for the top 50 most abundant miRNAs) were slightly more de-repressed than those of targets with a single Ago2 site (Supplementary Fig. 4e; $P = 0.07$; Wilcoxon’s rank sum test). The marginal P value may be because of the different data sources, which showed relatively high variation across replicates.”

To:

“The changes in expression of the targets with an Ago2-UPF1 overlapping site (for the 50 most abundant miRNAs) were marginally greater than those of the targets with a single Ago2 site (Supplementary Fig. 4e; $P = 0.07$; Wilcoxon’s rank-sum test).”

5. In Figures 4, 6 and 7 the authors are using 9 putative miR-24-3p targets. However, it is not shown in the manuscript that these are indeed miR-24-3p direct targets. This should be performed to substantiate the interpretation of the results shown in Figures 4, 6 and 7.

(Response) As shown in Fig. 4d, we used a miR-24-3p inhibitor to block miR-24-3p targeting, leading to derepression of nine putative targets and two known targets. We then introduced both the inhibitor and a miR-24-3p pseudo mimic (retaining a potency of base pairing to 7-mer sites but avoiding inhibition by the inhibitor) and observed repression of the targets (returning to the level of the control). We believe that these results indicate that the tested genes are potential miR-24-3p targets. However, to support our results, we also tested the miR-24-3p targets by mutagenizing the predicted miRNA binding sites in reporter constructs in the presence or absence of a miR-24-3p pseudomimic or miR-24-3p inhibitor, confirming that all the targets, except *PAK4*, were indeed direct targets of miR-24-3p (Supplementary Fig. 5 in the revised manuscript). Briefly, the miR-24-3p pseudomimic reduced the levels of firefly luciferase (FLuc) transcripts containing miR-24-3p-binding sites but did not affect the FLuc transcripts containing mutant sequences in the 3'UTR. Consistent with this finding, the miR-24-3p inhibitor increased the levels of FLuc transcripts containing miR-24-3p-binding sites but did not have any effect on the transcripts containing the mutant sequence. Based on these results, we have added a few sentences in the revised manuscript as shown below.

“To determine whether the putative miR-24-3p transcripts were directly regulated by miR-24-3p, the same mutagenesis was performed in the reporter constructs in either the presence or absence of a miR-24-3p pseudomimic or inhibitor (Supplementary Fig. 5). The RT-qPCR results suggested that the miR-24-3p pseudomimic reduced the levels of the WT reporter transcripts (except *PAK4*) but not those of the Mut transcripts. In contrast, the miR-24-3p inhibitor increased the levels of the WT reporter transcripts (except *PAK4*) but not those of the Mut transcripts, suggesting that the putative miR-24-3p targets were directly regulated by miR-24-3p.”

6. Along the same line, miRNA inhibitors against miR-24-3p and miR-26a are used (Figures 4 and Supplementary S4), however the significance of the inhibitor effect on the putative target expression is not shown. For most cases it appears to be a minor effect.

(Response) The significance of the inhibitory effects on the putative target expression is shown in Fig. 4 and Supplementary Fig. 4 and is indicated as “*”, $P < 0.05$. Because miRNA inhibitors are known to be degraded faster than locked nucleic acid (LNA) inhibitors in cells, if we had used the LNA inhibitors, we could have obtained more robust results. In addition, many laboratories have used miRNA inhibitors and have reported the relative ranges of derepression to be between 1.2- and 6.0-fold, as shown below. Furthermore, we employed the MXI1 and ATG4A transcripts as positive controls, and in previous reports, the level of ATG4A increased by approximately 3-fold in small cell lung cancer cells. In contrast, the derepression level in our study was 1.2-fold, which is the level normally observed. Thus, we strongly believe that the derepression levels were low but statistically significant.

Articles	miRNAs	Transcripts	Fold changes in target levels due to the miRNA inhibitors
Eichner et al. (2010) miR-378* Mediates Metabolic Shift in Breast Cancer Cells via the PGC-1b/ERRg Transcriptional Pathway, Cell Metabolism 12; 352–361	miR-378*	ESRRG , GABPA	1.2
Poliseno et al. (2010) A coding-independent function of gene and pseudogene mRNAs regulates tumour biology, Nature 465; 1033-1040	miR-17, miR-19	PTENP1	1.25, 1.5
Yu et al. (2017) Fusobacterium nucleatum Promotes Chemoresistance to Colorectal Cancer by Modulating Autophagy, Cell 170; 548-563	miR-18a*, miR-4802	ULK1 , ATG7	2.0
Xu et al. (2013) Derepression of a Neuronal Inhibitor due to miRNA Dysregulation in a Schizophrenia-Related Microdeletion, Cell 152; 262-275	miR-185	2310044H10Rik (Mirta22)	1.6
Li et al. (2018) Transcriptional Regulation of the Warburg Effect in Cancer by SIX1 , Cancer Cell 33;1-18	miR-548a-3p	SIX1	2.0, 6.0

7. Clarify the use of ‘pseudo-mimic strategy’ instead of classical miRNA mimics. This is particularly important since it has been thoroughly shown that the base pairing outside of the seed-sequence is important for target regulation.

(Response) Thank you for the comment. Although base pairing outside of the seed sequence (termed supplementary pairing) could have effects on miRNA targeting, many previous studies have shown that supplementary pairing has only a marginal effect but that the 7-mer sites, including a seed site, are major determinants for miRNA-mediated regulation of targets⁴⁻⁶. Based on this information, we designed a pseudo mimic that retained base pairing with a seed site but had a mutated outside sequence to avoid inhibition by miRNA inhibitors. To clarify the pseudo mimic strategy, we have added a description of the strategy in the revised manuscript, as shown below.

“A *pseudomimic* was used because *pseudo mimics* can form base pairs with the 7-mer sites of targets but avoid inhibition by *miR-inhibitors*” (page 13 line 263).

8. Indeed, the experiments using miRNA mimics and inhibitors are shown in rather complicated ways. An extreme example are the results shown in Figure 4F using a double normalization, resulting in difficult evaluation and interpretation of the results.

(Response) As per the reviewer’s suggestion, the results in Fig. 4f and S4i with double normalization were reproduced with less normalized or prenormalized results. The less processed results for Fig. 4f and S4i are provided in the “Reviewer only version” shown below. Briefly, the miRNA target levels were upregulated upon UPF1 depletion, and treatment with an miRNA inhibitor reduced the miRNA target transcript levels. These results are shown in Fig. 4b and 4d. Downregulation of UPF1 and treatment with an miRNA inhibitor increased the miRNA target transcript levels. To determine the effects of UPF1 on miRNA targets, we tried to compare the effects of UPF1 depletion in the presence or absence of the miRNA inhibitor. However, the

unprocessed results were more difficult to interpret than intended. Thus, we determined the ratios of the transcript levels under UPF1 depletion in the presence or absence of the miRNA inhibitor, as shown in Fig. 4f, demonstrating that the effects of UPF1 depletion were masked by the miRNA inhibitor. We agree with the reviewer's comment; however, we believe that the present, more processed, results will be more useful to the readers. If the reviewer suggests that we include the unprocessed results in the manuscript, we will include the "Reviewer only version" shown below.

Corresponding to Fig. 4f

Corresponding to Fig. S4i

9. The efficiency of TNRC6A/C knockdown should be demonstrated by Western-blotting.

(Response) We appreciate the reviewer's comment. We performed Western blotting to verify the efficiency of TNRC6A/C knockdown and observed efficient downregulation, as described in the revised Fig. 7c.

Revised Fig. 7c.

10. Along this line, positive controls for TNRC6A/C knockdown in terms of target repression should be shown using canonical miRNA targets (Ago2/TNRC6-dependent, UPF1-independent).

(Response) Thank you for this comment. We selected several genes, including TNGLN, S1PR2, LOXL2 and STK36 (miR-24-3p targets; UPF independent but TNRC6 dependent), as positive controls, and these genes exhibited derepression in TNRC6-depleted cells but not UPF1-depleted cells, as expected (Fig. 7d).

Revised Fig. 7d.

11. It would be most interesting and significantly strengthen the manuscript to demonstrate that, in the context of UMD, SMG7 is indeed recruiting the deadenylase complex to mediate miRNA-mediated decay.

(Response) Thank you for the comments. In addition to the co-IP of UPF1 and SMG7 with FLAG-AGO2, the AGO-UPF1-SMG7 axis is supported by the fact that siSMG7 conditions phenocopy those of siUPF1 in terms of miRNA-dependent UPF1-mediated regulation in the original study (Fig. 6). To strengthen our hypothesis, we determined whether FLAG-AGO2 could interact with the deadenylase complex via the UPF1-SMG7 complex. NOT1 and NOT3, components of the deadenylase complex, were coimmunoprecipitated using anti-FLAG-antibody in the presence or absence of UPF1. We found that FLAG-AGO2-NOT1/NOT3 physically interacted via UPF1, and this result has been described in Fig. 7b (in the revised manuscript).

Revised Fig. 7b.

To further confirm the mechanistic link between UPF1-SMG7 and AGO2-miRNA-mediated mRNA regulation, we constructed a SMG7 deletion mutant that could not bind to the deadenylase complex¹ and tested our mechanistic model (the AGO2-UPF1-SMG7-deadenylase complex). Briefly, we overexpressed wild-type FLAG-tagged SMG7 and the SMG7 mutant and quantified the UMD targets listed in Fig. 4 and 7. The RT-qPCR results indicated that SMG7 WT downregulated the miR-24-3p targets, but the SMG7 mutant did not have any effect on the miR-24-3p targets (Supplementary Fig. 7a and 7b in the revised manuscript). We hope that the reviewer will find that these new results to strongly support our mechanistic model.

Revised Supplementary Fig. 7.

12. Clarify the reason of the use of the term 'endogenous miRNAs' throughout the manuscript.

(Response) We used the term 'endogenous miRNAs' to differentiate between these miRNAs and the miRNA mimic. However, as all the miRNAs were endogenous, we have changed the term to 'miRNAs' throughout the manuscript.

Reviewer #3 (Remarks to the Author):

This study indicates that miRNAs containing CAG triplets within seeds may regulate targets through an alternative miRNA-mediated repressive pathway, which is dependent from UPF1/SMG7 but independent from TNRC6. In general, I believe that this is a highly important claim, but it is not yet well supported experimentally and bioinformatically. I am a bit puzzled regarding the control transcript sets used during the bioinformatic analyses and this should be better defined and clarified. Also, this study would significantly benefit from the use of CRISPR-CAS9 KO lines, to conclusively exclude siRNA-off target issues.

Specifically:

1) Page 6, lines 98 to 103. This part is unclear, the authors should explain it better. Which databases were used exactly here? It lacks of a small introduction and therefore it is arduous to understand. Difficult to comment on this part

(Response) Thank you for the comment. To describe the rationales and aims of the computational approach for selecting genuine non-NMD genes, we have added a few introductory descriptions and have revised the sentences as shown below. In addition, the database that we used is described in the "Data source" section in the Materials and Methods, and the detailed method for selecting non-NMD genes is described in the "PTC-free genes" section in the Materials and Methods.

From:

"It is well-known that UPF1 regulates the stability of mRNAs through a long 3'UTR. However, UPF1-dependent targets may include actual dEJs or PTC because the search for dEJs or alternative splicing isoforms was conducted using only a public database rather than by comprehensively analyzing actual dEJs or PTCs. We thus sought to exclude not only known NMD targets, but also all genes with potential PTCs to analyze the expression changes in PTC-free targets in UPF1-depleted cells."

To:

"Although early studies reported that a considerable fraction of RNAs responsive to UPF1 depletion are dependent on the 3'UTR length, the question of whether UPF1-responsive RNAs actually decay in an NMD-independent manner has been repeatedly asked. To address this unanswered question, we sought to identify genuine non-NMD genes and reanalyze these genes to check whether they continued to exhibit responses in UPF1-depleted cells. To identify non-NMD genes, we first removed all the genes that included any transcript with potential PTCs in the annotation (NCBI RefSeq: see the "Data Source" section of Materials and Methods for additional details) or PTCs in transcripts that were newly assembled from the RNA-seq data of UPF1-depleted cells using our computational pipeline (Fig. 1a; see the "PTC-free genes" and "Filtering genes with nonproductive splicing forms" sections in the Materials and Methods for additional details)."

2) It is unclear why replication-dependent histone genes should be excluded from the analysis, can this be clarified?

(Response) Thank you for the comment. Because some replication-dependent histone genes have been reported to be rapidly degraded by nuclear cap-binding proteins (CBPs) that are coupled to NMD during DNA replication⁷, we originally excluded these genes from the analysis. However, upon examining the changes in expression of the replication-dependent histone genes, the expression of a majority of these genes was not changed or decreased in UPF1-depleted cells (data not shown). We thus omitted this step in the revised computational pipeline for selecting PTC-free genes (revised Fig. 1a).

3) The authors should explain in details the strategy used to remove off-targets of siRNAs against UPF1, GFP and scrambled siRNAs. They refer to their "computational pipeline" that I could not find anywhere nor in the results section neither in the material and methods.

Response) Thank you for the comment. We removed all siRNA off-targets that included 7-mer sites (miRNA-like target sites: 8-mer, 7-mer m8, and 7-mer A1) in the 3'UTRs and described the method in the "Filtering siRNA off-targets" section of the Materials and Methods. To clarify the description in the main text, we have revised it as shown below.

From: *"We additionally removed all siRNA off-targets (for UPF1, GFP, and scrambled siRNAs) through our*

computational pipeline (Fig. 1a; see the Materials and Methods section for more details),”

To: “In addition, we removed all the siRNA off-targets (for UPF1, GFP, and the scrambled siRNAs) that included 7-mer sites in the 3’UTR (see “Filtering siRNA off-targets” section in the Materials and Methods for additional details)”

In the “Filtering siRNA off-targets” section of the Materials and Methods, we have added the detailed method as shown below.

“Because siRNA off-targeting is mediated by a partial complementarity between siRNA and 3’UTRs, similar to miRNA targeting, the same criteria used for miRNA 7-mer target sites were used to detect siRNA off-targets. mRNAs with 8mer, 7m8, or 7A1 site(s) of an siRNA in the 3’UTR were regarded as putative siRNA off-targets. Although not all putative off-targets are true off-targets, these mRNAs were rigorously filtered.”

4) In addition, I do not believe that removing off target bioinformatically is the correct step here because this procedure is based on prediction only. I would suggest to the authors to additionally perform their own RNA-seq experiments from CRISPR-UPF1-KO and control lines, also.

(Response) We agree that the computational prediction of off-targets is not a perfect approach. However, many previous studies have shown that siRNA off-targeting effects can be mostly explained by miRNA-like targeting via seed-based 7-mer sites⁸⁻¹⁰. We thus sought to filter off-targets that included 7-mer sites of siRNAs.

Knockout by CRISPR/Cas9 is performed via nonsense-mediated decay (NMD) of target RNAs that include indel-derived PTCs. Moreover, if the factors involved in NMD or the posttranscriptional regulatory mechanism were downregulated during the selection period, the effect of the KO would not be observed. Indeed, we generated a heterogeneous UPF1 knockout (KO) HeLa cell line in which multiple exons were deleted using paired gRNAs and CRISPR/Cas9 and performed RNA-seq analysis, but the results using the KO cell line were not consistent with what we and other groups had observed previously^{3,11}. The UMD targets observed in the UPF1-knockdown cells tended to be repressed rather than derepressed in the heterozygous KO cell line. We suggested that this discrepancy originated from the accumulation and secondary effects of a disruptive posttranscriptional regulation mechanism over long periods (approximately 3 months for the selection of clones). Thus, we strongly believe that transient downregulation of proteins by siRNA would be better for this study.

In addition, we analyzed the CDFs of PTC-free genes using publicly available data obtained by other laboratories (Fig. 1 and Supplementary Fig. 1), where different siUPF1 sequences were employed. As described in the manuscript, the CDF results from other laboratories were identical to our observations. To confirm that there were no off-target effects, we analyzed RNA-seq data for a siSMG7-treated HeLa cell line¹², for which a different siSMG7 sequence was employed. Indeed, SMG7 depletion increased the stability of transcripts harboring long 3’UTRs (Revised Supplementary Fig. 6e), suggesting that our observations were not the results of off-target effects.

Revised Supplementary Fig. 6e

We also included a description of the additional analysis in the revised manuscript as follows:

“To rule out whether the results obtained for siSMG7 were due to off-target effects of siRNAs and determine whether SMG7 is involved in UMD, we also analyzed other publicly available RNA-seq data for SMG7-depleted cell lines, for which different siSMG7 sequences were employed¹². Indeed, SMG7 depletion consistently increased the stability of transcripts harboring long 3’UTRs (Supplementary Fig. 6e), suggesting that our observation was not the result of off-target effects” (page 17 line 360).

5) It is unclear why the authors choose a cut-off of log₂ fold-change = 0.2 to select UPF1 targets. This filter looks too relaxed to me and could include a large number of off-targets.

(Response) Thank you for the comment. We decided to use the cut-off of log₂(x) = 0.2 (1.14 for fold change), similar to the median change for miRNA targets embedding a 7A1 site in miRNA-transfected cells (Nam, 2014 #5), allowing us to analyze UPF1-dependent miRNA targeting at the single-site level. Although the cut-off allows some off-targets in our analyses, we believe that the off-target effects were minimized by filtering siRNA off-targets.

6) The authors should also perform RNA-seq experiments using double KO or Kd lines for UPF1 and another component of the canonical NMD pathway (i.e. SMG6) to conclusively demonstrate their hypotheses.

(Response) We have shown that SMG5 and 6 are not associated with UPF1-dependent miRNA targeting in the siSMG5 and siSMG6 datasets. We do not believe that the double KO or KD of UPF1 and SMG5 or 6 can conclusively demonstrate our hypotheses. Furthermore, CRISPR/Cas9-based knock-out (KO) relies on the NMD mechanism. Thus, KO by CRISPR/Cas9 could not be achieved without NMD factors, including UPF1, SMG5 and SMG6. Additionally, HeLa cells have multiple chromosomes, which prevents complete deletion of nucleotides in all genes. We hope that this explanation will be acceptable to the reviewer.

7) Figure 2a. The authors highlight highest ranked 7-mers in grey which show a specific motif predominately starting with C or T, as evidenced by the motif logo on top of the graph. It is not clear from the manuscript the significance of this aspect.

(Response) Thank you for this comment. Our intended meaning was that the highly ranked 7-mers shown in gray tend to have high GC content. Instead of the entire sequence, we have inserted bar graphs that depict the GC content of the remaining highly ranked 7-mers and others in the revised Fig. 2a, as shown below.

8) Line 187-193, Figure 3a. It is unclear which transcripts are used as control here and the number of transcripts involved in the control signature. How could the authors select transcripts that lack endogenous 7-mer sites but contain random 7-mer sites? Transcripts lacking 7-mer sites are theoretically rare. Is the number of these transcripts enough for robust statistical testing? Also, why this analysis was not performed using cumulative distribution function?

(Response) Thank you for the comment. Briefly, the random controls are nontargets that do not contain 6-mer sites of the 50 most abundant miRNAs but contain 7-mer sites of 50 random miRNAs. Out of

thousands of PTC-free mRNAs, 275 were chosen as nontargets, of which more than 200 mRNAs with 7-mer sites of 50 random miRNAs were finally selected as random controls. As noted by the reviewer, because many mRNAs have endogenous 6-mer sites in the 3'UTRs, the random control candidates were reduced to a much lower number. However, this number was sufficient to test the statistical significance. For the statistical test, 1000 random control cohorts (each cohort included targets that had the 7-mer sites of the 50 random miRNAs) were generated, and the mean values of the random cohorts were compared to calculate the *P* value using one sample t-test. For improved clarity, we have described the method for generating random controls in the “Generation of random controls and statistical tests” section in Materials and Methods.

In addition, Fig. 3a depicts bar graphs instead of CDFs, as we needed to draw multiple CDF graphs, as shown below (Fig. B; “Reviewer only” version).

Reviewer only version; Fig. B.

9) The RT-qPCR experiment in Supplemental Figure 3c lacks statistical test because probably performed only once. In addition, the one in Supplemental Figure 3d is not convincing because log2 fold-change is too low. This cannot conclusively demonstrate that Dicer1 was kept downregulated for long enough to give the possibility to miRNAs, which tend to be particularly stable, to be down-regulated during this experiment. I would suggest the use of CRISPR-CAS9 Dicer1 KO lines, here.

(Response) Thank you for the comment. For Supplementary Fig. 3c, we have added replicates and indicated the *P*-values obtained using a t-test. Based on this result, the changes shown in Supplementary Fig. 3c did not result from decreased stability of RNAi. Because a single miRNA 7-mer site has an efficacy that can downregulate targets by log2 values from 0.2 to 0.5 on average in miRNA-transfected cells², the results in Supplementary Fig. 3d are acceptable when considering the remaining Dicer1 level. Regarding the generation of a Dicer1 KO cell line by CRISPR/Cas9, we did not try to make this cell line because the removal or depletion of Dicer1 with many cell passages over time (normally 3 months), we believe, will have accumulated secondary effects. We hope that our explanation is acceptable.

Revised Supplementary Fig. 3c.

10) Line 196. Also this experiment should be repeated with CRISPR-CAS9 lines.

(Response) We partially agree with the reviewer’s point that a double KO cell line (UPF1 and Dicer) generated by CRISPR/Cas9 will provide more convincing results. However, the generation of a double KO

cell line, including UPF1 KO, will be extremely difficult because the deletion or downregulation of UPF1, which is one of the factors of NMD, will reduce the NMD effects, which are ultimately required for CRISPR/Cas9. Furthermore, multiple nonhomologous end-joining events during the generation of a double KO cell line may cause disruptive DNA rearrangement, and multiple gRNAs could generate gRNA-derived off-targets. We hope that our explanation is acceptable.

11) In lines 244 to 248 the authors try to select UPF1-dependent miR-24-3p targets considering transcripts with miR-24-3p 8-mer or 7m8 mers, but no sites for other miRNAs. I looked at those transcripts in TargetScan and they actually appear to contain plenty of sites for other miRNAs, which are also conserved across species and repeated multiple times within the 3'UTRs (see for example miR-29-3p for ATAD2B, miRNA that does not contain CAG within its seed) indicative of active repression. The author should demonstrate that these transcripts are exclusively targeted by miR-24-3p, but not other miRNAs to consider this gene signature appropriate for following analyses. Same thing is valid for miR-26a

(Response) We selected miR-24-3p targets that contain 7-mer sites but no sites for the top 10 abundant miRNAs. As other nontop 10 miRNAs, including miR-29-3p (14,786 RPM), are expressed at much lower levels than miR-24-3p (26,356 RPM) in HeLa cells, we originally thought that the effects of other miRNAs would be much lower than those of miR-24-3p. However, to determine whether the selected targets are directly regulated by miR-24-3p, we employed a reporter system containing miR-24-3p-binding sites or a mutated binding sequence in the presence or absence of a miR-24-3p pseudomimic or an inhibitor. Briefly, the miR-24-3p pseudo mimic reduced the level of all firefly luciferase (FLuc) transcripts containing miR-24-3p-binding sites except *PAK4* but did not affect the FLuc transcripts containing mutant sequences. Consistent with this finding, a miR-24-3p inhibitor increased the levels of FLuc transcripts containing miR-24-3p-binding sites but did not have any effects on the transcripts containing the mutant sequence. Based on these results, we have added a few sentences to the revised manuscript (Supplementary Fig. 5).

Revised Supplementary Fig. 5.

“To determine whether the putative miR-24-3p transcripts were directly regulated by miR-24-3p, the same mutagenesis was performed in the reporter constructs in either the presence or absence of a miR-24-3p pseudomimic or inhibitor (Supplementary Fig. 5). The RT-qPCR results suggested that the miR-24-3p pseudomimic reduced the levels of the WT reporter transcripts (except PAK4) but not those of the Mut transcripts. In contrast, the miR-24-3p inhibitor increased the levels of the WT reporter transcripts (except PAK4) but not those of the Mut transcripts, suggesting that the putative miR-24-3p targets were directly regulated by miR-24-3p.”

12) To conclusively demonstrate that TNRC6 is not involved in this miRNA-repressive pathway the authors should perform RNA-seq from TNRC6 CRISPR-CAS9 stable KO cellular clones.

(Response) We agree with the reviewer's point that the TNRC6 KO cell line will give us more reliable results. However, for the same reasons described above, we strongly believe that the generation of the TNRC6 KO cell line over a long period will lead to accumulated effects of non-miRNA functions (secondary effects), resulting in the disruption of posttranscriptional regulation. Thus, transient downregulation of TNRC6 would be beneficial for the study of this mechanism. We hope that our explanation is appropriate and acceptable.

Minor points

The authors should remove endo-miRNAs throughout the manuscript and replace this term with miRNAs, because contrarily to siRNAs, miRNAs are endogenous by definition.

(Response) As per the reviewer's suggestion, we have changed this term to 'miRNAs' throughout the manuscript.

References

1. Loh, B., Jonas, S. & Izaurralde, E. The SMG5-SMG7 heterodimer directly recruits the CCR4-NOT deadenylase complex to mRNAs containing nonsense codons via interaction with POP2. *Genes Dev* 27, 2125-38 (2013).
2. Nam, J.W. et al. Global analyses of the effect of different cellular contexts on microRNA targeting. *Mol Cell* 53, 1031-43 (2014).
3. Hurt, J.A., Robertson, A.D. & Burge, C.B. Global analyses of UPF1 binding and function reveal expanded scope of nonsense-mediated mRNA decay. *Genome Res* 23, 1636-50 (2013).
4. Agarwal, V., Bell, G.W., Nam, J.W. & Bartel, D.P. Predicting effective microRNA target sites in mammalian mRNAs. *Elife* 4(2015).
5. Baek, D. et al. The impact of microRNAs on protein output. *Nature* 455, 64-71 (2008).
6. Bartel, D.P. Metazoan MicroRNAs. *Cell* 173, 20-51 (2018).
7. Choe, J. et al. Rapid degradation of replication-dependent histone mRNAs largely occurs on mRNAs bound by nuclear cap-binding proteins 80 and 20. *Nucleic Acids Res* 41, 1307-18 (2013).
8. Garcia, D.M. et al. Weak seed-pairing stability and high target-site abundance decrease the proficiency of Isy-6 and other microRNAs. *Nat Struct Mol Biol* 18, 1139-46 (2011).
9. Jackson, A.L. & Linsley, P.S. Recognizing and avoiding siRNA off-target effects for target identification and therapeutic application. *Nat Rev Drug Discov* 9, 57-67 (2010).
10. Riba, A. et al. Explicit Modeling of siRNA-Dependent On- and Off-Target Repression Improves the Interpretation of Screening Results. *Cell Syst* 4, 182-193 e4 (2017).
11. Imamachi, N., Salam, K.A., Suzuki, Y. & Akimitsu, N. A GC-rich sequence feature in the 3' UTR directs UPF1-dependent mRNA decay in mammalian cells. *Genome Res* (2016).
12. Colombo, M., Karousis, E.D., Bourquin, J., Bruggmann, R. & Muhlemann, O. Transcriptome-wide identification of NMD-targeted human mRNAs reveals extensive redundancy between SMG6- and SMG7-mediated degradation pathways. *RNA* 23, 189-201 (2017).

Reviewers' comments:

Reviewer #1 (Remarks to the Author):

I went through the revised version of the manuscript and must acknowledge that the authors have put a lot of effort and work into their study. While overall the presented data is suggestive for the existence of a UPF1/SMG7-dependent miRNA-mediated mRNA decay (which they call UMD), it is still not entirely compelling. However, for claiming the discovery of a new miRNA-mediated mRNA decay mechanism, the supporting evidence should be more compelling.

With all their data, the authors cannot rule out that they look at the additive effect of two unrelated but already well known mRNA decay pathways, NMD and conventional miRNA-mediated mRNA decay. Their most compelling evidence for a mechanistic link between UPF1 and miRNA-mediated mRNA decay is that a Dicer knockdown abrogates the UPF1-dependent downregulation on the so-called "3' UTR length-dependent PTC-free genes". However, this could also be an indirect effect since the UPF1 mRNA itself was shown to be regulated by miR-128 (Bruno et al., Mol Cell 2011). The experiments with miRNA inhibitors and pseudo-miRNA mimics in the context of UPF1 knockdowns (Fig. 4) are meant to document a mechanistic link, but the results the way they are displayed are hard to interpret and inconclusive in my opinion (see specific comments below). Finally, to rule out that we are not observing just a form of NMD on the interrogated genes, a high-quality mRNA-seq data set from cells with a SMG6 knockdown should be checked. Since the authors used the SMG7 knockdown data from Colombo et al. (RNA 2017), I suggest to also use the SMG6 knockdown data set from this study.

Specific comments to revised version:

Line 24/25: Please delete "respectively". The stability of mRNAs is also determined by NMD, not only by miRNAs.

There is a conceptual problem with the term "PTC-free genes": In addition to the general problem of the lack of a sound definition for the term "premature termination codon" (when shall we call it premature and when is it mature?), the ORF and hence its termination codon can only be defined at the level of a specific RNA transcript, dependent on the splicing of this transcript, but not at the level of the gene. This conceptual problem with the term PTC is also reflected in the confusing statement in the last paragraph of the discussion (lines 528-531), which should be deleted or changed. The authors claim that there are 4 subclasses of NMD and that three of them result from a PTC, but this is not correct. For example, mRNAs with an intron in the 3' UTR may well translate a full length ORF that codes for a functional protein, yet this mRNA can be targeted by NMD.

Line 138: in the same vein, what do the authors mean with "the canonical NMD pathway"? Since the NMD pathway is not yet understood in great detail, the authors should define what they mean with this term. A pragmatic and operational definition that can be used for metazoan NMD would be mRNA degradation that depends on UPF1 as well as on SMG6 and/or SMG7. According to this definition, UMD as described here would count as NMD, as it seems to require UPF1 and SMG7.

Fig. 2b: Although apparently statistically significant, the claimed enrichment of CUG triplets in UMD (with 7-mer MREs) is minor, there are almost as many CUG triplets in non-UMD targets (with 7-mer MREs). Thus, this conclusion should be toned down.

Fig. 3a and d: The authors claim that dysregulation of UMD targets and their 3'UTR length-dependent repression disappeared in cells with Dicer knockdown (lines 212-213). However, the pattern of mean expression changes remains very similar. The main difference is that in Fig. 3d, the SEM of the random control is much larger than in Fig. 3a, rendering the differences in 3a statistically significant and those in 3d not significant. What is the reason for this large variation in the random control in 3d but not in 3a? I would expect the random controls to behave similarly in each condition.

Fig. 4b and line 258: The authors conclude that 8 of the 9 selected targets were consistently de-repressed upon UPF1 knockdown. To me it looks as if only 4 of the 9 were de-repressed >2-fold, while 3 were de-repressed marginally and 2 were not affected. I feel there is an overselling of minor differences in order to claim a strong correlation between UPF1 knockdown and upregulation of the selected mRNAs when in fact this correlation is doubtful. By splitting data that should be compared to each other into different panels in Fig. 4, it also hides the fact that there is quite some variation among the selected transcripts in how their RNA levels are affected by UPF1 knockdown, and by the application of miR-24 inhibitors and miR-24 pseudo mimics. The comparison that is lacking and that would be informative would be to show side-by-side how each transcript level is affected by i) UPF1 knockdown, by ii) UPF1 knockdown combined with adding the miR-24 inhibitor, and by iii) UPF1 knockdown, miR-24 inhibitor and miR-24 pseudo mimic. That would mean to combine the data from 4b and 4e in one panel and to add the condition with the miR-24 inhibitor added without simultaneous addition of the miR-24 pseudo mimic. Finally, I am also a bit suspicious about the small differences in relative mRNA levels detected in the different conditions and the apparently extraordinarily low variation among biological replicates, as judged by the small SEMs. From my long experience with measuring RNA levels by RT-qPCR, it is seldom possible to obtain a p-value < 0.05 if two values differ by less than 1.5-fold. In this manuscript, it seems that differences of only 20% were statistically significantly different.

Fig. 7: In the revised version, the partial loss of UPF1 and SMG7 in the Ago2 IP upon RNase digestion is now acknowledged, but the authors nevertheless still conclude that this would indicate a direct (or at least an RNA independent) interaction between these proteins. I disagree for the following reason: Ago2 and UPF1 seem to bind on mRNA closer to each other than does Ago2 and PABP, hence it is less likely under mild RNase digestion conditions that the RNase cuts inbetween Ago2 and UPF1 than that it cuts inbetween Ago2 and PABP. Therefore, PABP completely disappears from the precipitated material while UPF1 and SMG7 only partially disappear. Thus, my conclusion from your data is that Ago2 and UPF1/SMG7 both are bound to the same mRNA molecules but that they do not directly interact with each other.

Reviewer #2 (Remarks to the Author):

The authors have extensively revised the manuscript and appropriately addressed my concerns. The revised manuscript is much improved; the findings are interesting and the conclusions drawn are supported by the data presented.

Minor comment:

In respect to comment #2 of Reviewer 1 and my comment #8 it would be useful to introduce the non-normalized data as supplementary figure.

Reviewers' comments:

Reviewer #1 (Remarks to the Author):

I went through the revised version of the manuscript and must acknowledge that the authors have put a lot of effort and work into their study. While overall the presented data is suggestive for the existence of a UPF1/SMG7-dependent miRNA-mediated mRNA decay (which they call UMD), it is still not entirely compelling. However, for claiming the discovery of a new miRNA-mediated mRNA decay mechanism, the supporting evidence should be more compelling. With all their data, the authors cannot rule out that they look at the additive effect of two unrelated but already well known mRNA decay pathways, NMD and conventional miRNA-mediated mRNA decay. Their most compelling evidence for a mechanistic link between UPF1 and miRNA-mediated mRNA decay is that a Dicer knockdown abrogates the UPF1-dependent downregulation on the so-called "3' UTR length-dependent PTC-free genes". However, this could also be an indirect effect since the UPF1 mRNA itself was shown to be regulated by miR-128 (Bruno et al., Mol Cell 2011). The experiments with miRNA inhibitors and pseudo-miRNA mimics in the context of UPF1 knockdowns (Fig. 4) are meant to document a mechanistic link, but the results the way they are displayed are hard to interpret and inconclusive in my opinion (see specific comments below). Finally, to rule out that we are not observing just a form of NMD on the interrogated genes, a high-quality mRNA-seq data set from cells with a SMG6 knockdown should be checked. Since the authors used the SMG7 knockdown data from Colombo et al. (RNA 2017), I suggest to also use the SMG6 knockdown data set from this study.

Thank you for the valuable comments and suggestions, which significantly improved our study. This reviewer requested more compelling evidence to support the presence of UPF1/SMG7-dependent miRNA-mediated mRNA decay; we considered the reason why you raised a concern about compelling evidence for UMD targeting might be partly because the results of the miRNA inhibitor and pseudo-mimic experiments were difficult to interpret. We have thus reproduced all results of the miRNA inhibitor and pseudo-mimic experiments in Fig. 4 after more efficiently knocking down UPF1 (Revised Fig. 4), and we have changed the representation of the results, as suggested, in the revised Supplementary Fig. 4f. In addition to the miRNA inhibitor and pseudo-mimic experiments, we also employed mutagenesis of the CUG motif in the miRNA target site, which showed that the UPF1-recognizing CUG motif is required for the additional effects of

miRNA targeting (Fig. 5). The UMD targets with a CUG motif in the miRNA target site showed a greater response to UPF1 knockdown (Fig. 6b, d), and miRNA targets with UMD signatures showed additional targeting in response to ectopic miRNA expression (Fig. 7i). We believe that all of these results demonstrate the presence of UMD targeting with an additive effect on miRNA-mediated mRNA decay, although we did not show how UPF1 and TNRC6 cooperatively interact with Ago2 from a mechanistic perspective.

This reviewer was also concerned that our results could be discounted by the fact that UPF1 may be regulated by miR-128. However, the downregulation of UPF1 or Dicer did not have any effects on the expression of the counterparts, as shown in Fig. 3b, and the downregulation of UPF1 did not significantly change the level of miRNAs in HeLa cells (Supplementary Fig. 3b). Furthermore, we found miR-128 to not be in the top 50 abundant miRNAs in HeLa cells. Thus, UPF1 does not seem to be regulated by miR-128.

As suggested, we also analyzed the SMG6 knockdown data reported by Colombo et al. (RNA 2017) and found that the PTC-free genes had little response to SMG6 RNAi but more response to SMG7 RNAi, as shown below.

Revised Supplementary Fig. 6e

Finally, to rule out siRNA-off target effects, we employed ASO-mediated downregulation, as shown in the revised Supplementary Fig. 9c-d. ASO-mediated downregulation confirmed all the results obtained by siRNA-mediated downregulation.

Supplementary Fig. 9c, d.

Thus, we hope you will now find our miRNA-mediated UMD model to be acceptable.

Specific comments to revised version:

Line 24/25: Please delete “respectively”. The stability of mRNAs is also determined by NMD, not only by miRNAs.

RE) Thank you for this comment. We have revised the manuscript accordingly.

There is a conceptual problem with the term “PTC-free genes”: In addition to the general problem of the lack of a sound definition for the term “premature termination codon” (when shall we call it premature and when is it mature?), the ORF and hence its termination codon can only be defined at the level of a specific RNA transcript, dependent on the splicing of this transcript, but not at the level of the gene. This conceptual problem with the term PTC is also reflected in the confusing statement in the last paragraph of the discussion (lines 528-531), which should be deleted or changed. The authors claim that there are 4 subclasses of NMD and that three of them result from a PTC, but this is not correct. For example, mRNAs with an intron in the 3' UTR may well translate a full length ORF that codes for a functional protein, yet this mRNA can be targeted by NMD.

RE) We appreciate this suggestion. By PTC-free genes, we meant those that

have no PTC in any transcript. Here, PTC was defined as a stop codon before the 50 nt upstream of the last exon junction. As mentioned, an mRNA with an intron in the 3'UTR can be targeted by NMD because the normal stop codon is considered a PTC by the splicing in the 3'UTR. Thus, we also excluded genes that have any splicing signal in the 3'UTR at the transcript level. Our PTC-free genes with long 3'UTRs are thus those with no splicing signal in the 3'UTR. However, we agree that our description of the PTC in the discussion could cause confusion, so we have revised the text, as follows.

“Our observations for both UPF1 and miRNA suggest an alternative gene silencing mechanism in the miRNA targeting pathway. Although we determined the new pathway, UMD, which is associated with the key NMD and miRNA pathway factors UPF1 and Ago2, understanding the detailed mechanisms after UPF1 and Ago2 complex formation and the functions of UMD in cell fate or the development of cancerous cells will provide new insights into posttranscriptional regulation.”

In addition, our definition of the premature termination codon redescribed in the Materials and Methods of the revised manuscript, as follows.

“Then, all potential NMD target genes, including putative PTCs (defined as a stop codon more than 50 nt upstream from the last exon junction), were excluded. The PTC can be generated by a cryptic splicing event in the 3'UTR, indicated by putative splicing signals (GU-AG with an inner distance greater than 60 nt) or with RNA-seq-supported exon junctions in the 3'UTR, and by nonproductive splicing patterns under UPF1-depleted conditions.” Page 28, lines 571.

Line 138: in the same vein, what do the authors mean with “the canonical NMD pathway”? Since the NMD pathway is not yet understood in great detail, the authors should define what they mean with this term. A pragmatic and operational definition that can be used for metazoan NMD would be mRNA degradation that depends on UPF1 as well as on SMG6 and/or SMG7. According to this definition, UMD as described here would count as NMD, as it seems to require UPF1 and SMG7.

RE) Thank you for this suggestion. The canonical NMD pathway is considered PTC-triggered (EJC-dependent) NMD in our manuscript. Although UMD is dependent on UPF1 and SMG7, it is not triggered by PTC but rather by miRNA

targeting. This is why we claimed this is a novel type of mRNA decay. To clarify our intended meaning, we have revised “canonical NMD” to “PTC-triggered NMD.”

Fig. 2b: Although apparently statistically significant, the claimed enrichment of CUG triplets in UMD (with 7-mer MREs) is minor, there are almost as many CUG triplets in non-UMD targets (with 7-mer MREs). Thus, this conclusion should be toned down.

RE) Thank you for this suggestion. We have revised the description, as follows.

“The CUG motif was significantly enriched in the 5-nt upstream and downstream windows of the MREs (based on the 5'-end of the MREs) compared to randomized controls ($P < 2.20 \times 10^{-16}$; Fisher's exact test; Fig. 2b and Supplementary Fig. 2b). Although not as much as that of the UMD targets, the CUG motif of the non-UMD targets was also significantly enriched in the windows compared to the randomized control ($P < 2.20 \times 10^{-16}$; Fig. 2b). We thus hypothesized that enrichment of the CUG motif could result from enrichment of the CAG (base-pairing to the CUG) motif in seed miRNAs.”

Fig. 3a and d: The authors claim that dysregulation of UMD targets and their 3'UTR length-dependent repression disappeared in cells with Dicer knockdown (lines 212-213). However, the pattern of mean expression changes remains very similar. The main difference is that in Fig. 3d, the SEM of the random control is much larger than in Fig. 3a, rendering the differences in 3a statistically significant and those in 3d not significant. What is the reason for this large variation in the random control in 3d but not in 3a? I would expect the random controls to behave similarly in each condition.

RE) Thank you for this comment. Although the pattern of the mean expression change slightly remained with Dicer knockdown, the magnitude of the change significantly decreased. As miRNAs are still expressed at a low level even under Dicer depletion, the pattern is still detectable. In fact, the standard error of the mean of the random control is similar between Fig. 3a and d. As we think that the scale of the y-axis may have caused confusion, we rescaled the y-axis in Fig. 3d to match that in Fig. 3a, as shown below.

Fig. 3a and revised Fig. 3d.

Fig. 4b and line 258: The authors conclude that 8 of the 9 selected targets were consistently de-repressed upon UPF1 knockdown. To me it looks as if only 4 of the 9 were de-repressed >2-fold, while 3 were de-repressed marginally and 2 were not affected. I feel there is an overselling of minor differences in order to claim a strong correlation between UPF1 knockdown and upregulation of the selected mRNAs when in fact this correlation is doubtful. By splitting data that should be compared to each other into different panels in Fig. 4, it also hides the fact that there is quite some variation among the selected transcripts in how their RNA levels are affected by UPF1 knockdown, and by the application of miR-24 inhibitors and miR-24 pseudo mimics. The comparison that is lacking and that would be informative would be to show side-by-side how each transcript level is affected by i) UPF1 knockdown, by ii) UPF1 knockdown combined with adding the miR-24 inhibitor, and by iii) UPF1 knockdown, miR-24 inhibitor and miR-24 pseudo mimic. That would mean to combine the data from 4b and 4e in one panel and to add the condition with the miR-24 inhibitor added without simultaneous addition of the miR-24 pseudo mimic.

Finally, I am also a bit suspicious about the small differences in relative mRNA levels detected in the different conditions and the apparently extraordinarily low variation among biological replicates, as judged by the small SEMs. From my long experience with measuring RNA levels by RT-qPCR, it is seldom possible to obtain a p-value < 0.05 if two values differ by less than 1.5-fold. In this manuscript, it seems that differences of only 20% were statistically significantly different.

RE) Thank you for these valuable comments. We agree that minor differences in the transcript levels are shown in Fig. 4b. To further support these results, we

repeated the experiments represented in Fig. 4b and f in the first revised manuscript and combined the results. With more than six biological replicates, we increased the amount of UPF1 siRNA transfected into cells from a low passage number. Consequently, we were able to downregulate expression twice as effectively. The revised results are similar to the original results shown in Fig. 4b; however, the relative levels of *ELL* and *PAK4* transcripts increased to ~2-fold greater than the control level. For a better level comparison, we have provided the previous Fig. 4b and the revised Fig. 4b below.

Fig. 4b and revised Fig. 4b.

As suggested, we compared each transcript level with or without treatment with siUPF1 or the miR-24-3p pseudo-mimic or inhibitor, as shown in Fig. 4b and f (revised Supplementary Fig. 4f).

Revised supplementary Fig. 4f

Fig. 7: In the revised version, the partial loss of UPF1 and SMG7 in the Ago2 IP

upon RNase digestion is now acknowledged, but the authors nevertheless still conclude that this would indicate a direct (or at least an RNA independent) interaction between these proteins. I disagree for the following reason: Ago2 and UPF1 seem to bind on mRNA closer to each other than does Ago2 and PABP, hence it is less likely under mild RNase digestion conditions that the RNase cuts in between Ago2 and UPF1 than that it cuts in between Ago2 and PABP. Therefore, PABP completely disappears from the precipitated material while UPF1 and SMG7 only partially disappear. Thus, my conclusion from your data is that Ago2 and UPF1/SMG7 both are bound to the same mRNA molecules but that they do not directly interact with each other.

RE) We agree with these comments on the relationship between RNase digestion and the proximity of two proteins. We also did not conclude that Ago2 directly interacts with UPF1 based on the IP results shown in Fig. 7 in the original or the first revised manuscript. Our intended meaning was that Ago2 could interact with UPF1 directly or indirectly through either RNAs or other mediators and that SMG7 interacts with Ago2 through UPF1, as you noted. However, to more clearly deliver what we found, we have modified the descriptions in Fig. 7a and b and provided an additional discussion of the issue in the Discussion section. We hope the modified conclusion and discussion are acceptable.

From:

“UPF1 and SMG7 coimmunoprecipitated with FLAG-Ago2 in a partially RNA-independent manner, suggesting that SMG7 interacts with Ago2.”

To:

“UPF1 and SMG7 coimmunoprecipitated with FLAG-Ago2 in a partially RNA-independent manner, suggesting that SMG7 could form a complex with Ago2.”

Page 20, lines 411.

In the Discussion section, we included an additional description of the interpretation, as follows.

“Although our IP results suggest interactions between Ago2 and UPF1, we cannot conclude that Ago2 directly interacts with UPF1 because the interaction could result from the proximal interaction of two proteins via either RNA or other mediators. The physical interaction between Ago2 and UPF1 requires further validation in the UMD context.” Page 25, lines 507.

Reviewer #2 (Remarks to the Author):

The authors have extensively revised the manuscript and appropriately addressed my concerns. The revised manuscript is much improved; the findings are interesting and the conclusions drawn are supported by the data presented.

Minor comment:

In respect to comment #2 of Reviewer 1 and my comment #8 it would be useful to introduce the non-normalized data as supplementary figure.

RE) Thank you for this suggestion. We have included the non-normalized data as Supplementary Fig. 4f for miR-24-3p and Supplementary Fig. 4k for miR-26a.

Revised Supplementary Fig. 4f

Revised Supplementary Fig. 4k

Reviewer #3 (Remarks to the Author):

I can see that the article is now, overall significantly improved compared to the original version. Nevertheless, my main perplexity about this study still remains the fact that the authors claim important finding solely by using RNAi technologies that, as the authors themselves noticed, create huge amount of off target issues. Also, in their pipeline of bioinformatic removal of off target genes the authors did not take into account the effect of transcripts containing 6mer seeds, which can be substantial (see for example Garcia DM et al, Nat Struct Mol Biol, 2011).

Nowadays, several alternative technologies of gene silencing have been established and must be used in parallel to RNAi to conclusively prove findings. This is particular important for this study. Contrarily to what assert the authors in their response letter, CRISPR knock out (KO) strategies can be used for KO of NMD factors, too. **UPF1 CRISPR KO cell lines are commercially available, for example**. Also, I am surprised that the authors are not aware that CRISPR-based KO strategy can be performed avoiding the need of NMD factors for silencing (such as removing promoters or using homologous recombination approaches) on NMD genes, such as UPF1 or other genes.

Nevertheless, I agree that at this stage creating CRISPR KO stable clones could be time consuming. However, to avoid relaying solely on siRNA-based technologies for their conclusions the authors should perform some of the crucial experiments also using alternative strategies of gene silencing which occur in transient. These include, for example, the use of antisense oligo nucleotides (ASO) which depends on RNaseH cleavage or CRISPR interference (CRISPRi). These should be used in in parallel to siRNA-based strategies here.

Thank you for these constructive comments and suggestions. We believe that our study was significantly improved by addressing these concerns and suggestions. Regarding the siRNA off-target issues, we have checked the enrichments of 6-mer seed sites in the UMD targets and the UMD targeting effects after excluding 6-mer seed sites. We found that 6-mer seed sites did not affect our results or conclusions, as shown below.

Regarding the other reviewer's suggestion, we had previously generated heterogeneous UPF1 KO cell lines in which multiple exons were deleted using paired gRNAs and CRISPR/Cas9, and we performed RNA-seq. However, the results using the KO cell line did not show the 3'UTR-length dependent RNA decay pattern, which is not consistent with previous observations in UPF1

knockdown using RNAi (data not shown). This difference probably resulted from the accumulation and/or secondary effects of disruptive posttranscriptional regulatory mechanisms over the long clone selection period. Moreover, if the factors involved in NMD or posttranscriptional regulatory mechanisms were downregulated during the selection period, the UMD targeting effect would not be observed. Thus, we alternatively knocked down UPF1 and SMG7 using antisense oligonucleotides (ASOs) and performed RNA-seq, and the UMD targeting was similarly detected in the miRNA-dependent manner. We hope our additional results and revised manuscript are acceptable.

Specifically:

1. On this new version of the manuscript the authors have finally revealed the strategy used to remove off-targets of siRNAs against UPF1, GFP and scramble siRNAs. As I anticipated above, the effect of 6mer seeds has not been considered in this study and this is crucial and should be performed. I consider, though, that removal of 6mer containing transcripts in addition to 7mers, could result in a small signature of transcript to be tested. However, the authors should at least verify and show with pictures that there is not enrichment 6mer containing transcripts after removal of 7mers containing transcripts for the siRNAs used, in siRNA treated versus control signatures.

RE) Thank you for this valuable suggestion. We agree with this concern regarding the off-target effects of 6-mer seed sites. As suggested, enrichment tests for all 6-mer seed sites of siRNAs in the 3'UTRs of the UMD targets upon UPF1 and SMG7 knockdown were performed, and all seed sites of the siRNAs (siUPF1, two siSMG7, and two siControl) except for one of siSMG7 were not significantly enriched compared to those of dinucleotide-shuffled random sequences, suggesting that our results were minimally or not affected by 6-mer off-target effects.

We then reanalyzed the 3'UTR-length-dependent RNA decay of PTC-free genes after excluding targets with 6-mer seed sites. As previously observed, the 3'UTR-length-dependent RNA decay by both UPF1 and SMG7 knockdown was still observed in the CDF graphs (Supplementary Fig. 9a).

Supplementary figure 9a.

We then tested whether the UMD targeting depends on the miRNA-mediated gene regulation even after excluding the 6-mer off-targets, and we still found similar miRNA-dependent UMD targeting (Supplementary Fig. 9b).

Supplementary Fig. 9b.

2. UPF1 should also be depleted transiently, using ASO or CRISPRi following RNA-seq. This should follow bioinformatic evaluation of all the various effects depicted in the study, that would derive by UPF1 depletion.

RE) Thank you for this valuable suggestion. As recommended, we performed UPF1 and SMG7 knockdown using ASOs, followed by RNA-seq. All bioinformatics analyses confirmed our main results, suggesting that our findings are not associated with off-target effects (Supplementary Fig. 9c, d). Briefly, we transiently transfected siRNAs or ASOs modified with phosphorothioate in the backbone and constrained methyl modifications at the 5' and 3' ends for UPF1 and SMG7; then, RNA-seq and western blotting were performed to quantify the

levels of target proteins and transcripts, as shown in Supplementary Fig. 9c, d. All UMD-related results and 7-mer motifs in the UMD targets were robustly observed.

Supplementary Fig. 9c, d.

We added these results to the Discussion section, as follows.

“Although the series of RNAi experiments for UPF1 and SMG7 supported the existence of UMD targeting, the results of our RNAi experiments could suffer from the issue of off-target effects. However, even after excluding all off-targets with 6-mer seed sites, our results and conclusions were not changed (Supplementary Fig. 9a, b), and repeated experiments using antisense oligonucleotides (ASOs), which depend on RNaseH cleavage, confirmed what we observed in our RNAi experiments (Supplementary Fig. 9c, d). These results suggest that our findings are not the results of off-target effects of siRNAs.” Page 24, lines 490.

3. Line 252. In my previous review I indicated that the authors try to select UPF1-dependent miR-24-3p targets considering transcripts with miR-24-3p 8-mer or 7m8 mers, but no sites for other miRNAs. As I mentioned before, the transcripts selected by the authors actually appear to contain plenty of sites for other miRNAs, which are also conserved across species and repeated multiple times within the 3’UTRs, indicating that they are actively repressed by other miRNAs. (see for example miR-29-3p for ATAD2B, miRNA that does not contain CAG within its seed). In this rebuttal letter, the authors justify their assertions,

indicating that miR-29b is not amongst the top 10 miRNAs, therefore cannot target ATAD2B, but this has not been experimentally demonstrated. However, the fact that miR-29b is not amongst the top 10 miRNAs expressed in HeLa cells cannot exclude that miR-29b does represses ATAD2B in HeLa cells. Notably, miR-29b is the 14th more expressed miRNAs in HeLa, as the author show in the manuscript, and the level of expression reach 14,786 RPM, which is a very high level and sufficient to actively repress transcripts. In addition, it has been previously demonstrated that miR-29b represses genes in HeLa cells. Therefore, the authors should at least remove the sentence in 256: “but no sites for other miRNAs”, in the revised version of the manuscript, because this is untrue.

RE) Thank you for these comments. We agree and have revised the sentence to “*but no sites for the other nine most abundant miRNAs*” in the revised manuscript.

4. Page 14, the sentence “Taken together, our findings suggest that UMD functions in miRNA-mediated regulation in addition with UPF1” should be re-written with more clarity.

RE) We have rephrased the sentence as shown below.

“*Taken together, our findings suggest that UMD functions via miRNA-mediated regulation.*”

5. Page 16, the new sentence from line 325 to 332 talks about reporter construct experiment, whereas the linked supplementary fig 5 indicates that this experiment involved measure of selected endogenous transcripts by RT-qPCR, instead of reporter. Could the authors clarify?

RE) Thank you for this point. The results shown in Supplementary Fig. 5 were obtained from the reporter assay in the presence or absence of the miR-24-3p pseudo-mimic. This misunderstanding could be the result of an incorrect figure legend because we described that the levels of the reporter transcripts were normalized to that of GAPDH mRNA. Thus, we revised the sentence from “*The level of mRNA was normalized to that of GAPDH mRNA*” to “*The level of FLuc mRNA was normalized to that of RLuc mRNA*”.

6. The authors overstate that UPF1-AGO2 regulates transcripts through a TNRC6 independent manner for several reasons:

- 1) The authors did not silence all the TNRC6 members in these experiments, but only TNRC6A and C. Therefore, transcripts that do not change following TNRC6A/C silencing could still depend on TNRC6B.
- 2) The authors demonstrate this aspect only indirectly, looking at change in mRNA levels following TNRC6A/C and/or UPF1 silencing.

Therefore, the authors should soften their conclusion on this particular aspect. They cannot be totally sure that TNRC6 is not involved in UPF1-dependent silencing by the experiments performed.

RE) Thank you for these valuable comments. We agree with these points. As we knocked down TNRC6A/C but not B, the tested transcripts could still depend on TNRC6B. When we originally tried to downregulate all TNRC6s using multiple siRNAs, the knockdown of all three members was not efficiently achieved, probably due to excessive siRNA transfection. We thus decided to downregulate two members. As suggested, we have revised our conclusion and statements related to the relationship between UMD and TNRC6 throughout the manuscript. We have also revised the title of the section from "*TNRC6-independent, UPF1-dependent miRNA targeting pathway*" to "*UPF1-dependent miRNA-targeting pathway does not seem to require TNRC6A/C*".

It would be interesting to evaluate whether depletion of UPF1 increases AGO2-TNRC6 interaction and vice versa depletion of TNRC6s increases AGO2-UPF1 interaction. This would increase the validity of their model.

RE) Thank you for these valuable suggestions for validating our model. We agree that additional experiments to determine TNRC6 and UPF1 to bind Ago2 may confirm a competitive or cooperative relationship between them. Because the main point of our study was to demonstrate the presence of UMD targeting depending on miRNA-mediated mRNA decay and NMD factors, studying the dynamic interactions (competitive or cooperative) among Ago2, UPF1 and TNRC6 is beyond our scope of this study. However, it would be interesting to further examine the molecular dynamics between UPF1/Ago2 and TNRC6/Ago2 in a future study.

Reviewers' comments:

Reviewer #1 (Remarks to the Author):

In this manuscript, the authors' main conclusion is that they have discovered a novel silencing pathway that they call UMD and which requires miRNAs, AGO2, UPF1 and SMG7, but not TNRC6. For this claim to hold, the authors must be able to unambiguously demonstrate that what they measure is not merely the combined effect of two already well-studied mRNA decay pathways, nonsense-mediated and miRNA-mediated mRNA decay (NMD and miR-MD, respectively), but indeed a third distinct pathway. As I outlined in my previous comments and also here below, the presented data altogether are suggestive - yet not compelling - for the existence of such a novel pathway. For none of the reported observations, it can be ruled out that it does not represent the mixed effect of NMD and miR-MD, in which case the novelty of the study would be reduced to showing that UPF1 appears to preferentially bind 3'UTRs at CUG motifs. Methodologically, the problem is that the tools used to inactivate NMD (siRNAs) and miR-MD (ASO inhibitors) only lead to an incomplete inactivation, and under such conditions it is impossible to perform a classical epistasis analysis that would allow to unambiguously distinguish whether one is dealing with two separate or only one decay pathway. Since UPF1 knockouts seem not possible (lethal for cells according to our lab's experience [I have strong doubts that the commercially available UPF1 CRISPR KO cells mentioned by reviewer 3 are really full functional knockouts]), the most compelling yet not absolutely perfect experiment would be to mutate the miRNA binding sites in 3' UTRs of transcripts that are claimed to be subjected to UMD. If UMD indeed existed, UPF1 knockdown should not increase the RNA level of these constructs anymore.

In the present form, the results are not convincing enough to claim the discovery of a novel mRNA decay pathway

Specific points:

The authors are still operating with "PTC-free genes" to filter their data, which is a problematic concept as described in my previous review. In their response, they define it as genes "that have no PTC in any transcript" and further define PTC "as a stop codon before 50 nt upstream of the last exon junction". The problem is that this all relies on database annotations, which are far from complete and in particular lack many aberrantly spliced transcript versions of a gene, many of which tend to be targeted by NMD. In addition, their PTC definition includes many stop codons that are by no means premature and hence the term PTC as used by the authors in this manuscript is misleading.

Lines 138-140: Changing the criticized term "canonical NMD pathway" to "PTC-triggered NMD pathway" makes this statement trivial, and superfluous, because the data was initially filtered to only include "PTC-free" genes. So of course, whatever observation you make with these "PTC-free" genes should not be triggered by a PTC. Again, and there may be a semantics element to this, there is evidence for many PTC-free transcripts being targeted by NMD.

This reviewer still does not understand the rationale for mixing the miR-24-3p inhibitor with the miR-24-3p pseudo mimic in Fig. 4e. As shown in Fig. 4d, the combination of the two cancel out each other and hence the changes we see upon UPF1 knockdown in Fig. 4e are attributed to a miRNA-independent effect, i.e. to NMD rather than UMD. It is therefore also not surprising that the data looks very similar than the data in Fig. 4b.

On lines 284/5, the authors state that "our findings suggest that UPF1 and miRNA function in an additive manner". I agree and would be tempted to conclude that these results are consistent with two independent pathways. In theory, if UPF1 and the miRNA would function in the same pathway, the inactivation of each of them individually should have the same effect as inactivating them both together. In reality however, since we are dealing here with incomplete inactivations (knockdown,

inhibitors), the double inactivation might actually give the bigger effect than the inactivation of each factor individually. In conclusion, this data remains inconclusive with regards to whether we are observing one or two independent pathways.

Fig. 5: Since the CUG motifs might indeed represent the preferred binding sites for UPF1, the mutation of the CUGs that are part of the miR24 binding sites might affect both NMD and miR-MD, and hence this experiment also does not help to distinguish between the postulated new UMD model and the less exciting (because already known) scenario of NMD and miR-MD affecting the transcript independently. Instead, miRNA binding sites should be mutated in a way that is not likely to also affect UPF1 binding.

Fig. 7: This reviewer still disagrees with the interpretation of the co-IP data. While the authors take the observed reduction of UPF1 and SMG7 co-IP with Flag-AGO2 upon RNaseA treatment as an argument to suggest these factors interact in a partially RNA-independent manner, I would conclude that the association between UPF1/SMG7 and AGO2 is primarily bridged by RNA and the reason for not observing a complete loss upon RNaseA treatment is incomplete RNA digestion, leaving some shorter pieces of RNA intact. Association would then still be detected in the cases where UPF1 and AGO2 are bound to an RNA molecule close to each other.

Reviewer #2 (Remarks to the Author):

My comments had been addressed appropriately in the first round of revision. I consider that the findings reported are interesting and the conclusions drawn are supported by the data presented.

Reviewer #3 (Remarks to the Author):

I do not have any further comment/remark

Commenting on R#1 concerns:

I'm still not entirely convinced about the model proposed by the authors in this study. In cells, UTRs are usually regulated by several miRNAs at the same time. Therefore, how does the transcript being regulated select the mechanism of repression to utilise (canonical or new 'UMD') when interacts with multiple miRNAs?

I think it would be a good idea to use CRISPR to remove the miRNA binding sites in 3' UTRs of transcripts that are claimed to be subjected to UMD. If UMD indeed existed, UPF1 knockdown should not increase the RNA level of these constructs anymore", as suggested by R#1

Reviewers' comments:

Reviewer #1 (Remarks to the Author):

In this manuscript, the authors' main conclusion is that they have discovered a novel silencing pathway that they call UMD and which requires miRNAs, AGO2, UPF1 and SMG7, but not TNRC6. For this claim to hold, the authors must be able to unambiguously demonstrate that what they measure is not merely the combined effect of two already well-studied mRNA decay pathways, nonsense-mediated and miRNA-mediated mRNA decay (NMD and miR-MD, respectively), but indeed a third distinct pathway. As I outlined in my previous comments and also here below, the presented data altogether are suggestive - yet not compelling - for the existence of such a novel pathway. For none of the reported observations, it can be ruled out that it does not represent the mixed effect of NMD and miR-MD, in which case the novelty of the study would be reduced to showing that UPF1 appears to preferentially bind 3'UTRs at CUG motifs.

Methodologically, the problem is that the tools used to inactivate NMD (siRNAs) and miR-MD (ASO inhibitors) only lead to an incomplete inactivation, and under such conditions it is impossible to perform a classical epistasis analysis that would allow to unambiguously distinguish whether one is dealing with two separate or only one decay pathway. Since UPF1 knockouts seem not possible (lethal for cells according to our lab's experience [I have strong doubts that the commercially available UPF1 CRISPR KO cells mentioned by reviewer 3 are really full functional knockouts]), the most compelling yet not absolutely perfect experiment would be to mutate the miRNA binding sites in 3' UTRs of transcripts that are claimed to be subjected to UMD. If UMD indeed existed, UPF1 knockdown should not increase the RNA level of these constructs anymore.

In the present form, the results are not convincing enough to claim the discovery of a novel mRNA decay pathway

Response) Thank you for the constructive comments regarding compelling evidence for the existence of the UMD pathway. Although we demonstrated that UMD requires miRNA seed-type targeting and CUG triplet nucleotides within the vicinity of 7-mer sites and our observations did not result from off-targeting effects of siRNAs by presenting reproducible results with different siRNA sequences and ASOs, as the reviewer mentioned, the incomplete inactivation of factors might still raise a concern that the UMD pathway may be a simple combination of two well-known mRNA decay pathways, miRNA-mediated mRNA decay and NMD. In Fig. 5f-i, we show that UMD requires the CUG triplet nucleotide in 7-mer sites and that miRNA targeting with mutated CUG in the sites was no longer affected by UPF1 knockdown. However, this result is also true if the CUG triplet solely affects UMD without miRNA targeting, even though ablation of miRNA targeting also impaired UMD, as shown in Fig. 5h. Hence, to investigate whether the CUG triplet is solely sufficient enough for UMD, regardless of miRNA targeting, we mutated 7-mer sites except for the CUG in the reporter constructs and examined changes in expression of the reporters (Fig. 5j in the revised manuscript). Inactivation of miRNA targeting still impaired UMD regardless of the CUG triplet nucleotides, indicating that UMD requires miRNA targeting in a CUG-dependent manner and that UMD is not merely a simple combination of NMD and miRNA targeting.

Fig. 5j in the revised manuscript.

Specific points:

The authors are still operating with “PTC-free genes” to filter their data, which is a problematic concept as described in my previous review. In their response, they define it as genes “that have no PTC in any transcript” and further define PTC “as a stop codon before 50 nt upstream of the last exon junction”. The problem is that this all relies on database annotations, which are far from complete and in particular lack many aberrantly spliced transcript versions of a gene, many of which tend to be targeted by NMD. In addition, their PTC definition includes many stop codons that are by no means premature and hence the term PTC as used by the authors in this manuscript is misleading.

Response) Thank you for the comment. As the reviewer mentioned, the gene annotation is far from complete and lacks many aberrantly spliced transcripts destabilized by NMD. Thus, in examining cells, we also performed transcriptome assemblies to identify aberrantly spliced transcripts in siUPF1-treated cells and then excluded genes with any aberrant transcripts. To clarify our procedure, we have updated Fig. 1a and Materials and Methods section to properly demonstrate how we identified transcripts with dEJ and aberrantly spliced transcripts not listed in annotations. We agree that our definition of a PTC could include normal stop codons. To exclude EJC-dependent NMD targets in our analyses, we originally defined a PTC as a stop codon before 50 nt upstream of the last exon junction. However, in the revised manuscript, to avoid confusion, we describe EJC-dependent NMD targets as transcripts with downstream exon junctions (dEJ; defined as “the exon junctions after 50 nt downstream of the stop codon”) to remove EJC-dependent NMD targets. The remainder were considered dEJ-free genes subjected to downstream analyses.

Fig. 1a. in the revised manuscript

Lines 138-140: Changing the criticized term “canonical NMD pathway” to “PTC-triggered NMD pathway” makes this statement trivial, and superfluous, because the data was initially filtered to only include “PTC-free” genes. So of course, whatever observation you make with these “PTC-free” genes should not be triggered by a PTC. Again, and there may be a semantics element to this, there is evidence for many PTC-free transcripts being targeted by NMD.

Response) We agree with the reviewer’s point. Because we hypothesized that the 3’UTR-length-dependent regulation of UPF1-dependent mRNA decay (UMD) is related to the EJC-independent NMD pathway, we excluded EJC-dependent NMD target mRNAs, as described above, and included dEJ-free genes in downstream analyses. We examined whether 3’UTR-length-dependent regulation is still detectable for dEJ-free genes. Throughout the revised manuscript, we changed the term “PTC-triggered NMD pathway” to “EJC-dependent NMD pathway” and the term “PTC-free genes” to “dEJ-free genes”.

As shown in Fig. 4d, the combination of the two cancel out each other and hence the changes we see upon UPF1 knockdown in Fig. 4e are attributed to a miRNA-independent effect, i.e. to NMD rather than UMD. It is therefore also not surprising that the data looks very similar than the data in Fig. 4b.

Response) The rationale for the experiments presented in Fig. 4e was to confirm whether miRNA targeting via “seed-pairing” is actually required for UMD because the pseudo-mimic is not inhibited by the miR-24-3p inhibitor, unlike miR-24-3p, and base-pairs with targets via only 7-mer sites. Thus, the combination of miR-24-3p inhibitor and pseudo-mimic does not cancel out each one. However, because the conclusion for this experiment can also be derived from Fig. 4b and d, we have removed Fig 4e from the revised version to avoid unnecessary confusion.

On lines 284/5, the authors state that “our findings suggest that UPF1 and miRNA function in an additive manner”. I agree and would be tempted to conclude that these results are consistent with two independent pathways. In theory, if UPF1 and the miRNA would function in the same pathway, the inactivation of each of them individually should have the same effect as inactivating them both together. In reality however, since we are dealing here with incomplete inactivations (knockdown, inhibitors), the double inactivation might actually give the bigger effect than the inactivation of each factor individually. In conclusion, this data remains inconclusive with regards to whether we are observing one or two independent pathways.

Response) Our intended meaning of the additive effect was that the canonical miRNA targeting pathway (via TNRC6) and our new mRNA decay pathway (via UPF1/SMG7 but not TNRC6A/C) can be added to our results. However, because this appears to be an overstatement without results later in the manuscript, we removed the sentence.

Fig. 5: Since the CUG motifs might indeed represent the preferred binding sites for UPF1, the mutation of the CUGs that are part of the miR24 binding sites might affect both NMD and miR-MD, and hence this experiment also does not help to distinguish between the postulated new UMD model and the less exciting (because already known) scenario of NMD and miR-MD affecting the transcript independently. Instead, miRNA binding sites should be mutated in a way that is not likely to also affect UPF1 binding.

As mentioned above, to examine whether our observations are due to the activity of two independent pathways, additional mutagenesis in miRNA 7-mer sites but remaining CUG triplets in them was performed, revealing that miRNA targeting through CUG-embedding 7-mer sites is required for UMD. The results of the mutagenesis experiment of the CUG motif shown in Fig. 7i and this experiment suggest that UMD is not simply a simple combination of two independent pathways but rather a distinct RNA decay pathway that requires both UPF1 and miRNA targeting. To clarify our results, we added a description of the results, as follows:

“Because a miRNA inhibitor may not completely remove miR-24-3p, the UPF1-independent function via the CUG motif might also explain our qRT-PCR results. To exclude the possibility of the UPF1-independent function via the CUG motif, we mutated the 7-mer site with the remaining CUG (CUGAGCC to CUGCCAG) in the reporter transcripts (Mut-B). The level of the Mut-B transcripts was not affected by downregulating UPF1 (Fig. 5j), suggesting that UMD triggers mRNA decay via miRNA-mediated gene regulation in a CUG-motif-dependent manner and that the cooperative roles of Ago2 and UPF1 may be necessary for UMD.” lines 319-325.

Fig. 5j in the revised manuscript.

Fig. 7: While the authors take the observed reduction of UPF1 and SMG7 co-IP with Flag-AGO2 upon RNaseA treatment as an argument to suggest these factors interact in a partially RNA-independent manner, I would conclude that the association between UPF1/SMG7 and AGO2 is primarily bridged by RNA and the reason for not observing a complete loss upon RNaseA treatment is incomplete RNA digestion, leaving some shorter pieces of RNA intact. Association would then still be detected in the cases where UPF1 and AGO2 are bound to an RNA molecule close to each other.

Response) As the reviewer mentioned, the interaction of Ago2 and UPF1/SMG7 may still be RNA dependent due to incomplete digestion of RNAs or be captured via other mediators, despite RNase treatment. Because the binding sites of Ago2 and UPF1 in the 3'UTRs greatly overlap each other (Supplementary fig. 5c and d), they appear to be at least co-localized together. In that sense, we have restated that UPF1/SMG7 interacts with the RNA-induced silencing complex (RISC), including Ago2, rather than directly interacting with each other. To better clarify our intended meaning, we revised the subtitle of the section and the discussion, as follows:

“UPF1-SMG7 forms a complex with Ago2. To determine whether the UPF1-SMG7 complex is associated with the Ago2 complex, HeLa cells were transiently transfected with FLAG-Ago2 or empty vector containing FLAG-tag.”, lines 386-388.

and

“Although our IP results suggest interactions between Ago2 and UPF1, we cannot completely exclude the possibility that Ago2 indirectly interacts with UPF1 because the

interaction may result from the proximal interaction of two proteins via either undigested RNA or other mediators.”, lines 486-489.

Reviewer #2 (Remarks to the Author):

My comments had been addressed appropriately in the first round of revision. I consider that the findings reported are interesting and the conclusions drawn are supported by the data presented.

Reviewer #3 (Remarks to the Author):

I do not have any further comment/remark

Commenting on R#1 concerns:

I'm still not entirely convinced about the model proposed by the authors in this study. In cells, UTRs are usually regulated by several miRNAs at the same time. Therefore, how does the transcript being regulated select the mechanism of repression to utilise (canonical or new 'UMD') when interacts with multiple miRNAs?

I think it would be a good idea to use CRISPR to remove the miRNA binding sites in 3' UTRs of transcripts that are claimed to be subjected to UMD. If UMD indeed existed, UPF1 knockdown should not increase the RNA level of these constructs anymore”, as suggested by R#1

According to Reviewer #1's comment and our response, additional mutagenesis in miRNA 7-mer sites but remaining CUG triplets in them was performed to examine whether our observations are due to two independent pathways. The results indicated that miRNA targeting through CUG-embedding 7-mer sites is required for UMD. The results of mutagenesis of the CUG motif shown in Fig 7i and this experiment suggest that UMD is not merely a simple combination of two independent pathways but rather a distinct RNA decay pathway that requires both UPF1 and miRNA targeting. To clarify our results, we added a description of the results, as follows:

“Because a miRNA inhibitor may not completely remove miR-24-3p, the UPF1-independent function via the CUG motif might also explain our qRT-PCR results. To exclude the possibility of the UPF1-independent function via the CUG motif, we mutated the 7-mer site with the remaining CUG (CUGAGCC to CUGCCAG) in the reporter transcripts (Mut-B). The level of the Mut-B transcripts was not affected by downregulating UPF1 (Fig. 5j), suggesting that UMD triggers mRNA decay via miRNA-mediated gene regulation in a CUG-motif-dependent manner and that the cooperative roles of Ago2 and UPF1 may be necessary for UMD.” lines 319-325.

Fig. 5j in the revised manuscript.